# Self-Supervised Disentanglement by Leveraging Structure in Data Augmentations

## Abstract

Self-supervised representation learning often uses data augmentations to induce some invariance to "style" attributes of the data. However, with downstream tasks generally unknown at training time, it is difficult to deduce *a priori* which attributes of the data are indeed "style" and can be safely discarded. To deal with this, current approaches try to retain some style information by tuning the degree of invariance to some particular task, such as ImageNet object classification. However, prior work has shown that such task-specific tuning can lead to significant performance degradation on other tasks that rely on the discarded style. To address this, we introduce a more principled approach that seeks to *disentangle* style features rather than discard them. The key idea is to add multiple style embedding spaces where: (i) each is invariant to *all-but-one* augmentation; and (ii) *joint* entropy is maximized. We formalize our structured data-augmentation procedure from a causal latent-variable-model perspective, and prove identifiability of both content *and* individual style variables. We empirically demonstrate the benefits of our approach on both synthetic and real-world data.

## 1 Introduction

Learning useful representations from unlabelled data is widely recognized as an important step towards more capable machine-learning systems (Bengio et al., 2013). In recent years, *self-supervised learning* (SSL) has made significant progress towards this goal, approaching the performance of supervised methods on many downstream tasks (Ericsson et al., 2021). The main idea is to leverage known data structures to construct proxy tasks or objectives that act as a form of (self-)supervision. This could involve predicting one part of an observation from another (Brown et al., 2020), or, as we focus on in this work, leveraging **data augmentations/transformations** to perturb different attributes of the data.

Most current approaches are based on the joint-embedding framework and use data augmentations as weak supervision to determine what information to retain (termed "content") and what information to discard (termed "style") (Bromley et al., 1994; Chen et al., 2020a; Zbontar et al., 2021; Bardes et al., 2022). In particular, they do so by optimizing for representation similarity or **invariance** across transformations of the same observation, subject to some form of **entropy** regularization, with this invariance-entropy trade-off tuned for some particular task (e.g., ImageNet object classification).

However, at pre-training time, it is unclear what information should be discarded, as **one task's style may be another's content**. Ericsson et al. (2021) illustrated this point, finding ImageNet object-classification accuracy (the task optimized for in pre-training) to be poorly correlated with downstream object-detection and dense-prediction tasks, concluding that *"universal pre-training is still unsolved"*.

> **Example 1.1** (Color and Rotation)**.** Suppose we want to make use of color and rotation transformations. While some invariance to (or discarding of) an image's color and orientation features can be *beneficial* for ImageNet object classification (Chen et al., 2020a), it can also be *detrimental* for other tasks like segmentation or fine-grained species classification (Cole et al., 2022).

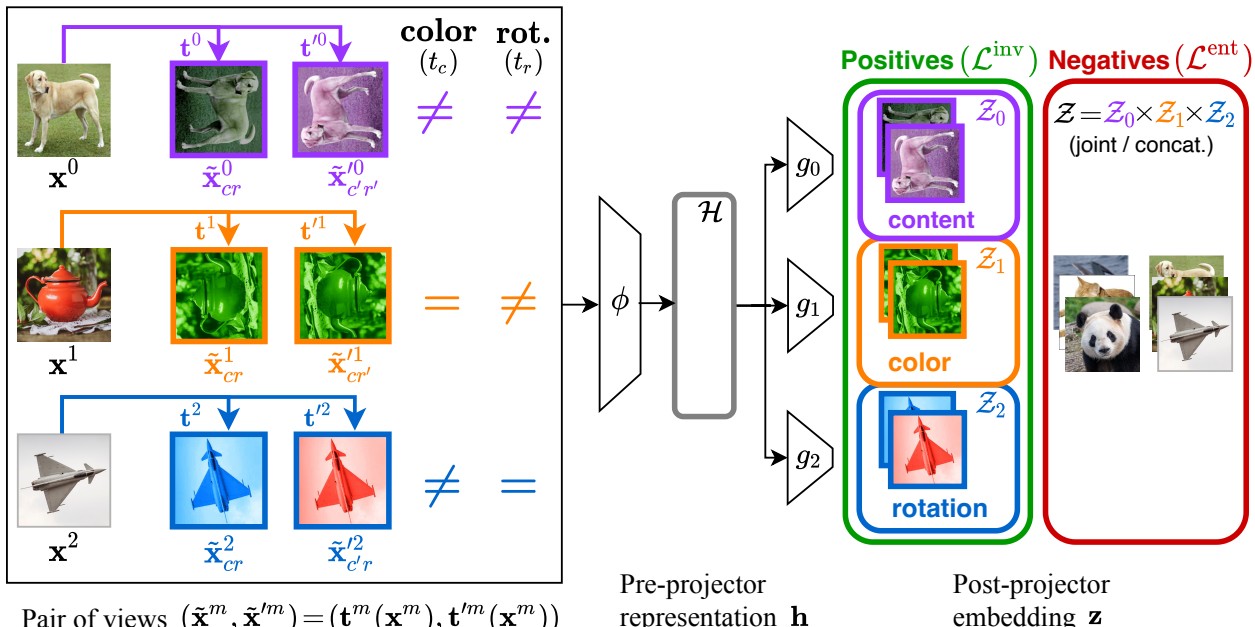

Figure 1: **Framework overview.** Given $M$ atomic transformations like color distortion or rotation (here, $M=2$), we learn a "content" embedding space ($\mathcal{Z}_0$) that is invariant to *all* transformations and $M$ "style" embedding spaces ($\mathcal{Z}_1$, $\mathcal{Z}_2$) that are each invariant to *all-but-the-$m^{th}$* atomic transformation. To do so, we construct $M+1$ transformation pairs ($\mathbf{t}^m, \mathbf{t}'^m$) *sharing different transformation parameters* and use these to create $M+1$ transformed image pairs ($\tilde{\mathbf{x}}^m, \tilde{\mathbf{x}}'^m$) *sharing different features*. After routing each pair to a different space, we: (i) enforce *invariance within* each space; and (ii) maximize *entropy across* the joint spaces. The result is $M+1$ *disentangled* embedding spaces.

To address this and learn more universal representations, we introduce a new SSL framework which uses data augmentations to **disentangle style attributes of the data rather than discard them**. As illustrated in Fig. 1, we leverage $M$ transformations to learn $M+1$ *disentangled* embedding spaces capturing both content and style information—with one style space per (group of) transformation(s).

**Structure and contributions.** The rest of this paper is organized as follows. We first provide some background on common SSL objectives in Section 2, and discuss how they seek to discard style information from the embedding space. In Section 3, we describe our framework for using data augmentations to disentangle and keep style information, rather than discard it. Next, in Section 4, we analyse our framework through the lens of causal representation learning, proving identifiability of both content *and* style variables. In our experiments of Section 5, we first use synthetic data to show how content can be *completely* disentangled from style, and then use ImageNet to show how, by keeping more style information, our framework improves downstream performance with the post-projector embedding. We end with a discussion of related work in Section 6 before concluding in Section 7. Our main contributions can be summarized as follows:

- **Algorithmic:** We propose a self-supervised framework for using data augmentations to *disentangle and keep* style information, rather than discard it (Section 3).

- **Theoretical:** We formalize our framework from a causal latent-variable-model perspective and prove identifiability of both content *and* individual style variables (Section 4).

- **Experimental:** We show: (i) how to *completely* disentangle content from style (Sections 5.1 and 5.2); and (ii) that our framework improves downstream performance with the post-projector embedding $\mathbf{z}$ by retaining more style information (Section 5.3); and (iii) that a large performance disparity remains between $\mathbf{z}$ and the pre-projector representation $\mathbf{h}$ (see Fig. 1), raising further questions about the role of the projector in SSL—particularly in regard to style retention (Section 5.3).

Table 1: **Common SSL Objectives via Invariance and Entropy.** Many SSL methods can be expressed as a (weighted) combination of **invariance** $\mathcal{L}^{\text{inv}}$ and **entropy** $\mathcal{L}^{\text{ent}}$ terms. Notation: $\mathbf{Z} = [\mathbf{z}^1, \mathbf{z}^2, \ldots, \mathbf{z}^n]$ and $\mathbf{Z}' = [\mathbf{z}'^1, \mathbf{z}'^2, \ldots, \mathbf{z}'^n]$ are batches of $n$ vectors of $d$-dimensional representations with $\mathbf{Z}, \mathbf{Z}' \in \mathbb{R}^{n \times d}$; $\mathbf{Z}_{\cdot j} \in \mathbb{R}^n$ is a vector composed of the values at dimension $j$ for all $n$ vectors in $\mathbf{Z}$; $\text{Cov}(\mathbf{Z}) = 1/(n-1) \sum_i (\mathbf{z}^i - \bar{\mathbf{z}})(\mathbf{z}^i - \bar{\mathbf{z}})^\top$ is the *sample* covariance matrix of $\mathbf{Z}$ with $\bar{\mathbf{z}} = 1/n \sum_{i=1}^n \mathbf{z}^i$; $\lambda_v$ and $\lambda_c$ are hyperparameters for weighting the variance and covariance terms, respectively; and $\epsilon > 0$ is a small scalar preventing numerical instabilities. ==Note that $n$ refers to the number of observations (images) in a batch==.

| Algorithm | $\mathcal{L}^{\text{inv}}(\mathbf{Z}, \mathbf{Z}')$ | $\mathcal{L}^{\text{ent}}(\mathbf{Z}, \mathbf{Z}')$ |
|---|---|---|
| SimCLR | $-\frac{1}{n} \sum_{i=1}^n \frac{(\mathbf{z}^i)^\top \mathbf{z}'^i}{\|\mathbf{z}^i\|\|\mathbf{z}'^i\|}$ | $\frac{1}{n} \sum_{i=1}^n \log \sum_{j=1, \neq i}^n \exp\left(\frac{(\mathbf{z}^i)^\top \mathbf{z}'^j}{\|\mathbf{z}^i\|\|\mathbf{z}'^j\|}\right)$ |
| BTs | $\sum_{j=1}^d \left(1 - \frac{(\mathbf{Z}_{\cdot j})^\top \mathbf{Z}'_{\cdot j}}{\|\mathbf{Z}_{\cdot j}\|\|\mathbf{Z}'_{\cdot j}\|}\right)^2$ | $\sum_{j=1}^d \sum_{k=1, \neq j}^d \left(\frac{(\mathbf{Z}_{\cdot j})^\top \mathbf{Z}'_{\cdot k}}{\|\mathbf{Z}_{\cdot j}\|\|\mathbf{Z}'_{\cdot k}\|}\right)^2$ |
| VICReg | $-\frac{1}{n} \sum_{i=1}^n \left\|\mathbf{z}^i - \mathbf{z}'^i\right\|_2^2$ | $\frac{\lambda_v}{d}\left(\sum_{j=1}^d \max\left(0, 1 - \sqrt{\text{Var}(\mathbf{Z}_{\cdot j}) + \epsilon}\right) + \max\left(0, 1 - \sqrt{\text{Var}(\mathbf{Z}'_{\cdot j}) + \epsilon}\right)\right) +$ $\frac{\lambda_c}{d}\left(\sum_{i=1}^n \sum_{j=1, \neq i}^n [\text{Cov}(\mathbf{Z})]_{i,j}^2 + [\text{Cov}(\mathbf{Z}')]_{i,j}^2\right)$ |

## 2 Background: Using *unstructured* data augmentations to discard

Joint-embedding methods are often categorized as contrastive or non-contrastive; while both employ some invariance criterion $\mathcal{L}^{\text{inv}}$ to encourage the same embedding across different views of the same image (e.g., cosine similarity or mean squared error), they differ in how they regularize this invariance criterion to avoid collapsed or trivial solutions. In particular, contrastive methods (Chen et al., 2020a;b; 2021; He et al., 2020) do so by pushing apart the embeddings of different images, while non-contrastive methods do so by architectural design (Grill et al., 2020; Chen and He, 2021) or by regularizing the covariance of embeddings (Zbontar et al., 2021; Bardes et al., 2022; Ermolov et al., 2021). We focus on *contrastive* and *covariance-based non-contrastive* methods which can both be expressed as a combination of **invariance** $\mathcal{L}^{\text{inv}}$ and **entropy** $\mathcal{L}^{\text{ent}}$ terms (Garrido et al., 2023),

$$\mathcal{L}^{\text{SSL}} = \mathcal{L}^{\text{inv}} + \mathcal{L}^{\text{ent}}. \tag{2.1}$$

Note these terms have also been called alignment and uniformity (Wang and Isola, 2020), respectively. For concreteness, Table 1 specifies $\mathcal{L}^{\text{inv}}$ and $\mathcal{L}^{\text{ent}}$ for some common SSL methods, namely SimCLR (Chen et al., 2020a), BarlowTwins (BTs, Zbontar et al. 2021), and VICReg (Bardes et al., 2022).

In general, the joint-embedding framework involves an unlabelled dataset of observations or images $\mathbf{x}$ and $M$ transformation distributions $\mathcal{T}_1, \ldots, \mathcal{T}_M$ from which to sample $M$ atomic transformations $t_1, \ldots, t_M$, with $t_m \sim \mathcal{T}_m$, composed together to form $\mathbf{t} = t_1 \circ \cdots \circ t_M$. Critically, *each atomic transformation $t_m$ is designed to perturb a different "style" attribute of the data* deemed nuisance for the task at hand. Returning to Example 1.1, this could mean sampling parameters for a color distortion $t_c \sim \mathcal{T}_c$ and rotation $t_r \sim \mathcal{T}_r$, and then composing them as $\mathbf{t} = t_c \circ t_r$. For brevity, this sample-and-compose operation is often written as $\mathbf{t} \sim \mathcal{T}$.

For each image $\mathbf{x}$, a pair of transformations $\mathbf{t}, \mathbf{t}' \sim p_{\mathbf{t}}$ is sampled and applied to form a pair of views $(\tilde{\mathbf{x}}, \tilde{\mathbf{x}}') = (\mathbf{t}(\mathbf{x}), \mathbf{t}'(\mathbf{x}))$. The views are then passed through a shared backbone network $\phi$ to form a pair of representations $(\mathbf{h}, \mathbf{h}')$, with $\mathbf{h} = \phi(\tilde{\mathbf{x}})$, and then through a smaller network or "projector" $g$ to form a pair of embeddings $(\mathbf{z}, \mathbf{z}')$, with $\mathbf{z} = g(\mathbf{h}) = g(\phi(\tilde{\mathbf{x}})) \in \mathcal{Z}$. Critically, the single embedding space $\mathcal{Z}$ seeks invariance to all transformations, thereby *discarding each of the "style" attributes.*

## 3 Our Framework: Using *structured* data augmentations to disentangle

We now describe our framework for using data augmentations to *disentangle* style attributes of the data, rather than discard them—see Fig. 1 for an illustration. Given $M$ transformations, we learn $M+1$ embedding spaces $\{\mathcal{Z}_m\}_{m=0}^M$ capturing both content ($\mathcal{Z}_0$) and style ($\{\mathcal{Z}_m\}_{m=1}^M$) information—with one style space per (group of) atomic transformation(s).

### 3.1 Views

We start by constructing $M+1$ transformation pairs $\{(\mathbf{t}^m, \mathbf{t}'^m)\}_{m=0}^M$ which *share different transformation parameters*. For $m=0$, we independently sample two transformations $\mathbf{t}^0, \mathbf{t}'^0 \sim \mathcal{T}$, which will generally[1] not share any transformation parameters (i.e., $t_k^0 \neq t_k'^0$ for all $k$). For $1 \leq m \leq M$, we also independently sample two transformations $\mathbf{t}^m, \mathbf{t}'^m \sim \mathcal{T}$, but then enforce that **the parameters of the $m^{\text{th}}$ transformation are shared** by setting $t_m'^m := t_m^m$. Finally, we apply each of these transformation pairs to a different image to form a pair of views $\left(\tilde{\mathbf{x}}^m, \tilde{\mathbf{x}}'^m\right) = \left(\mathbf{t}^m(\mathbf{x}^m), \mathbf{t}'^m(\mathbf{x}^m)\right)$.

> **Example 1.1 (continued).** Suppose we can sample parameters for two transformations: color distortion $t_c \sim \mathcal{T}_c$ and rotation $t_r \sim \mathcal{T}_r$. As depicted in Fig. 1, we can then construct three transformation pairs *sharing different parameters*: $(\mathbf{t}^0, \mathbf{t}'^0) = (t_c^0 \circ t_r^0, t_c'^0 \circ t_r'^0)$ with no shared parameters; $(\mathbf{t}^1, \mathbf{t}'^1) = (t_c^1 \circ t_r^1, t_c^1 \circ t_r'^1)$ with shared color parameters $t_c^1$; and $(\mathbf{t}^2, \mathbf{t}'^2) = (t_c^2 \circ t_r^2, t_c'^2 \circ t_r^2)$ with shared rotation parameters $t_r^2$. Applying each transformation pair to a different image, we get three pairs of views: $(\tilde{\mathbf{x}}_{cr}^0, \tilde{\mathbf{x}}_{c'r'}^0)$ for which only "content" information is shared as both color and rotation differ; $(\tilde{\mathbf{x}}_{cr}^1, \tilde{\mathbf{x}}_{cr'}^1)$ for which "content" and color information is shared, but rotation differs; and $(\tilde{\mathbf{x}}_{cr}^2, \tilde{\mathbf{x}}_{c'r}^2)$ for which "content" and rotation information is shared, but color differs.

### 3.2 Embedding spaces

As depicted in Fig. 1, the pairs of views $(\tilde{\mathbf{x}}^m, \tilde{\mathbf{x}}'^m)$ are passed through a shared backbone network $\phi$ to form pairs of representations $(\mathbf{h}^m, \mathbf{h}'^m)$ and subsequently through *separate* projectors $g_l$ to form pairs of embeddings $(\mathbf{z}_l^m, \mathbf{z}_l'^m)$, with

$$\mathbf{z}_l^m = g_l(\mathbf{h}^m) = g_l \circ \phi(\tilde{\mathbf{x}}^m) = g_l \circ \phi \circ \mathbf{t}^m(\mathbf{x}^m) \in \mathcal{Z}_l \tag{3.1}$$

being the embedding of view $\tilde{\mathbf{x}}^m$ in embedding space $\mathcal{Z}_l$. We call $\mathcal{Z}_0$ "content" space as it seeks invariance to *all* transformations, thereby discarding all style attributes and leaving only content. We call the other $M$ spaces $\{\mathcal{Z}_m\}_{m=1}^M$ "style" spaces as they seek invariance to *all-but-one* transformation $t_m$, thereby discarding *all-but-one* style attribute (that which is perturbed by $t_m$).

### 3.3 Loss

Given $M+1$ pairs of views $\{(\tilde{\mathbf{x}}^m, \tilde{\mathbf{x}}'^m)\}_{m=0}^M$ sharing different transformation parameters, we learn $M+1$ disentangled embedding spaces by minimizing the following objective:

$$\mathcal{L}^{\text{ours}}\left(\phi, \{g_m\}_{m=0}^M; \{(\tilde{\mathbf{x}}^m, \tilde{\mathbf{x}}'^m)\}_{m=0}^M\right) = \underbrace{\lambda_0 \mathcal{L}_{\mathcal{Z}_0}^{\text{inv}} + \mathcal{L}_{\mathcal{Z}_0}^{\text{ent}}}_{\text{standard loss (content} \to \mathcal{Z}_0)} + \underbrace{\left(\sum_{m=1}^M \lambda_m \mathcal{L}_{\mathcal{Z}_m}^{\text{inv}}\right) + \mathcal{L}_{\mathcal{Z}}^{\text{ent}}}_{\text{additional terms (style} \to \mathcal{Z}_m\text{'s})}, \tag{3.2}$$

$$= \underbrace{\left(\sum_{m=0}^M \lambda_m \mathcal{L}_{\mathcal{Z}_m}^{\text{inv}}\right)}_{M+1 \text{ inv. terms}} + \underbrace{\mathcal{L}_{\mathcal{Z}}^{\text{ent}}}_{\text{joint entropy}} + \underbrace{\mathcal{L}_{\mathcal{Z}_0}^{\text{ent}}}_{\text{content entropy}}, \tag{3.3}$$

where the individual invariance ($\mathcal{L}_{\mathcal{Z}_m}^{\text{inv}}$) and (content / joint) entropy ($\mathcal{L}_{\mathcal{Z}_0}^{\text{ent}}$ / $\mathcal{L}_{\mathcal{Z}}^{\text{ent}}$) terms are given by

$$\mathcal{L}_{\mathcal{Z}_m}^{\text{inv}} = \mathcal{L}^{\text{inv}}\left(\mathbf{z}_m^m, \mathbf{z}_m'^m\right), \qquad \mathcal{L}_{\mathcal{Z}_0}^{\text{ent}} = \mathcal{L}^{\text{ent}}\left(\{\mathbf{z}_0^m, \mathbf{z}_0'^m\}_{m=0}^M\right), \qquad \mathcal{L}_{\mathcal{Z}}^{\text{ent}} = \mathcal{L}^{\text{ent}}\left(\{\mathbf{z}^m, \mathbf{z}'^m\}_{m=0}^M\right),$$

with $\mathbf{z}^m = [\mathbf{z}_0^m, \dots, \mathbf{z}_M^m] \in \mathcal{Z}_0 \times \dots \times \mathcal{Z}_M$ the concatenated embeddings of $\tilde{\mathbf{x}}^m$ across all spaces, and $\{\lambda_m\}_{m=0}^M$ are hyperparameters weighting the invariance terms. Note that most SSL methods employ similar hyperparameters for tuning their invariance-entropy trade-offs[2] (see Table 1). To allow any "base" algorithm to be

---

[1] Always true for continuous transformations, and often true for discrete transformations.
[2] e.g., BarlowTwins and VICReg have explicit $\lambda$'s, while SimCLR controls this trade-off implicitly via the temperature $\tau$.

used, we have defined our loss using general invariance and entropy terms, with concrete specifications for these given in Table 1 (note that $\mathcal{L}^{\text{ent}}\left(\{\mathbf{z}^m, \mathbf{z}'^m\}_{m=0}^M\right)$ could be written as $\mathcal{L}^{\text{ent}}(\mathbf{Z}, \mathbf{Z}')$, with $\mathbf{Z} = [\mathbf{z}^0, \mathbf{z}^1, \ldots, \mathbf{z}^M]$ and $\mathbf{Z}' = [\mathbf{z}'^0, \mathbf{z}'^1, \ldots, \mathbf{z}'^M]$, to better align our loss notation with that of Table 1). Here, Eq. (3.2) highlights the additional terms we add to the standard SSL loss of Eq. (2.1), while Eq. (3.3) highlights the fact that we require **two different entropy terms to ensure disentangled embedding spaces**. Since "content" is invariant to all transformations (by definition), we require $\mathcal{L}_{\mathcal{Z}}^{\text{ent}}$ to prevent redundancy ($M$ additional copies of content, one in each style space) and $\mathcal{L}_{\mathcal{Z}_0}^{\text{ent}}$ to ensure content is indeed encoded in $\mathcal{Z}_0$ (otherwise it could be spread across all $M+1$ spaces). As discussed in Section 6, this is a key difference compared to Xiao et al. (2021), who also learn multiple embedding spaces but do not achieve disentanglement. In particular, by using a per-embedding-space entropy term (rather than a joint term), Xiao et al. (2021) get a (tangled) copy of content in each style embedding space.

# 4 Causal Representation Learning Perspective and Identifiability Analysis

In this section, we investigate *what* is actually learned by the structured use of data augmentations in Section 3, through the lens of causal representation learning (Schölkopf et al., 2021). To this end, we first formalize the data generation and augmentation processes as a (causal) latent variable model, and then study the identifiability of different components of the latent representation. Our analysis strongly builds on and extends the work of von Kügelgen et al. (2021) by showing that the structure inherent to different augmentation transformations can be leveraged to identify not only the block of shared content variables, but also *individual style components* (subject to suitable assumptions).

## 4.1 Data-generation and augmentation processes

We assume that the observations $\mathbf{x} \in \mathcal{X}$ result from *underlying latent vectors* $\mathbf{z} \in \mathcal{Z}$ via an invertible *nonlinear mixing function* $f : \mathcal{Z} \to \mathcal{X}$,

$$\mathbf{z} \sim p_{\mathbf{z}}, \qquad \mathbf{x} = f(\mathbf{z}). \tag{4.1}$$

Here, $\mathcal{Z} \subseteq \mathbb{R}^d$ is a *latent space* capturing object properties such as color or rotation; $p_{\mathbf{z}}$ is a distribution over latents; and $\mathcal{X}$ denotes the $d$-dimensional data manifold, which is typically embedded in a higher dimensional pixel space. In the same spirit, we model the way in which augmented views $(\tilde{\mathbf{x}}, \tilde{\mathbf{x}}')$ are generated from $\mathbf{x}$ through perturbations in the latent space:

$$\tilde{\mathbf{z}}, \tilde{\mathbf{z}}' \sim p_{\tilde{\mathbf{z}}|\mathbf{z}}, \qquad \tilde{\mathbf{x}} = f(\tilde{\mathbf{z}}), \qquad \tilde{\mathbf{x}}' = f(\tilde{\mathbf{z}}'). \tag{4.2}$$

The conditional $p_{\tilde{\mathbf{z}}|\mathbf{z}}$, from which the pair of *augmented latents* $(\tilde{\mathbf{z}}, \tilde{\mathbf{z}}')$ is drawn given the original latent $\mathbf{z}$, constitutes the latent-space analogue of the image-level transformations $(\mathbf{t}, \mathbf{t}') \sim \mathcal{T}$ in Section 3. More specifically, $\tilde{\mathbf{z}} \sim p_{\tilde{\mathbf{z}}|\mathbf{z}}$ captures the behavior of $f^{-1} \circ \mathbf{t} \circ f$ with $\mathbf{t} \sim \mathcal{T}$ acting on $\mathbf{x} = f(\mathbf{z})$.

## 4.2 Content-style partition

Typically, augmentations are designed to affect some semantic aspects of the data (e.g., color and rotation) and not others (e.g., object identity). We therefore partition the latents into *style* latents $\mathbf{s}$, which *are* affected by the augmentations, and shared *content* latents $\mathbf{c}$, which *are not* affected by the augmentations. Further, $p_{\tilde{\mathbf{z}}|\mathbf{z}}$ in (4.2) takes the form

$$p_{\tilde{\mathbf{z}}|\mathbf{z}}(\tilde{\mathbf{z}} \mid \mathbf{z}) = \delta(\tilde{\mathbf{c}} - \mathbf{c}) p_{\tilde{\mathbf{s}}|\mathbf{s}}(\tilde{\mathbf{s}} \mid \mathbf{s}) \tag{4.3}$$

for some style conditional $p_{\tilde{\mathbf{s}}|\mathbf{s}}$, such that $\mathbf{z}$, $\tilde{\mathbf{z}}$, and $\tilde{\mathbf{z}}'$ in (4.2) are given by

$$\mathbf{z} = (\mathbf{c}, \mathbf{s}), \qquad \tilde{\mathbf{z}} = (\mathbf{c}, \tilde{\mathbf{s}}), \qquad \tilde{\mathbf{z}}' = (\mathbf{c}, \tilde{\mathbf{s}}'). \tag{4.4}$$

For this setting, it has been shown that—under suitable additional assumptions—contrastive SSL recovers the shared content latents $\mathbf{c}$ up to an invertible function (von Kügelgen et al., 2021, Thm. 4.4).

### 4.3 Beyond content identifiability: separating and recovering individual style latents

Previous analyses of SSL with data augmentations considered style latents $\mathbf{s}$ as nuisance variables that should be discarded, thus seeking a pure content-based representation that is invariant to all augmentations (von Kügelgen et al., 2021; Daunhawer et al., 2023). The focus of our study, and its key difference to these previous analyses, is that *we seek to also identify and disentangle different style variables*, by leveraging available structure in data augmentations that has not been exploited thus far.

First, note that each class of atomic transformation $\mathcal{T}_m$ (e.g., color distortion or rotation) typically affects a different property, meaning that it should only affect a subset of style variables. Hence, we partition the style block into more fine-grained individual *style components* $s_m$,

$$\mathbf{s} = (s_1, ..., s_M), \qquad \tilde{\mathbf{s}} = (\tilde{s}_1, ..., \tilde{s}_M), \qquad \tilde{\mathbf{s}}' = (\tilde{s}'_1, ..., \tilde{s}'_M), \tag{4.5}$$

and assume that the style conditional $p_{\tilde{\mathbf{s}}|\mathbf{s}}$ in (4.3) factorizes as follows:

$$p_{\tilde{\mathbf{s}}|\mathbf{s}}(\tilde{\mathbf{s}} \mid \mathbf{s}) = \prod_{m=1}^{M} p_{\tilde{s}_m|s_m}(\tilde{s}_m \mid s_m), \tag{4.6}$$

where each term $p_{\tilde{s}_m|s_m}$ on the RHS is the latent-space analogue of $t_m \sim \mathcal{T}_m$.

Next, we wish for our latent variable model to capture the *structured* use of data augmentation through transformation pairs with *shared parameters*, as described in Section 3. Specifically, note that—unlike most prior approaches to SSL with data augmentations—we do *not* create a single dataset of ("positive") pairs $(\tilde{\mathbf{x}}, \tilde{\mathbf{x}}')$. Instead, we construct transformation pairs $(\mathbf{t}^m, \mathbf{t}'^m)$ in $M{+}1$ different ways, giving rise to $M{+}1$ datasets of pairs $(\tilde{\mathbf{x}}^m, \tilde{\mathbf{x}}'^m)$, each differing in the shared (style) properties. In particular, the $m^{\text{th}}$ atomic transformation is shared across $(\mathbf{t}^m, \mathbf{t}'^m)$ by construction, such that $(\tilde{\mathbf{x}}^m, \tilde{\mathbf{x}}'^m)$ should share the same perturbed $m^{\text{th}}$ style components $\tilde{s}_m = \tilde{s}'_m$—*regardless of its original value $s_m$*. To model this procedure, we define $M{+}1$ different ways of jointly perturbing the style variables as follows:

$$p^{(m)}(\tilde{\mathbf{s}}, \tilde{\mathbf{s}}' \mid \mathbf{s}) = \prod_{l=1}^{M} p^{(m)}(\tilde{s}_l, \tilde{s}'_l \mid s_l) \qquad \text{for} \qquad m = 0, \ldots, M,$$

$$\text{where} \qquad p^{(m)}(\tilde{s}_l, \tilde{s}'_l \mid s_l) = \begin{cases} p_{\tilde{s}_l|s_l}(\tilde{s}_l \mid s_l)\,\delta(\tilde{s}'_l - \tilde{s}_l) & \text{if} \quad l = m \\ p_{\tilde{s}_l|s_l}(\tilde{s}_l \mid s_l)\,p_{\tilde{s}_l|s_l}(\tilde{s}'_l \mid s_l) & \text{otherwise} \end{cases} \tag{4.7}$$

Together with $\mathbf{z} = (\mathbf{c}, \mathbf{s}) \sim p_{\mathbf{z}}$ as in (4.1), the conditionals in (4.7) induce $M{+}1$ different joint distributions $p^{(m)}_{\tilde{\mathbf{x}}, \tilde{\mathbf{x}}'}$ over observation pairs $(\tilde{\mathbf{x}}^m, \tilde{\mathbf{x}}'^m)$: analogous to (4.2), we have for $m = 0, \ldots, M$,

$$\tilde{\mathbf{s}}^m, \tilde{\mathbf{s}}'^m \sim p^{(m)}_{\tilde{\mathbf{s}}, \tilde{\mathbf{s}}'|\mathbf{s}}, \qquad \tilde{\mathbf{x}}^m = f\left([\mathbf{c}, \tilde{\mathbf{s}}^m]\right), \qquad \tilde{\mathbf{x}}'^m = f\left([\mathbf{c}, \tilde{\mathbf{s}}'^m]\right). \tag{4.8}$$

**Remark 4.1.** In practice, we do not generate $M{+}1$ augmented pairs for each $\mathbf{x} = f(\mathbf{z})$ as described above. Instead, each pair is constructed from a different observation with $\mathbf{x}^l = f(\mathbf{z}^l)$ transformed according to $m := l \mod M{+}1$. In the limit of infinite data, these two options have the same effect.

---

**Example 1.1 (continued).** Denote the style component capturing color by $s_c$ and that capturing rotation by $s_r$. For $m = 0, 1, 2$, let $\mathbf{z}^m = (\mathbf{c}^m, s_c^m, s_r^m)$ be the latents underlying separate images $\mathbf{x}^m$. Then the augmentations shown in Fig. 1 (left) are captured by the following changes to the latents:

| $m$ | $\mathbf{z}^m$ | $\tilde{\mathbf{z}}^m$ | $\tilde{\mathbf{z}}'^m$ | Shared Latents |
|---|---|---|---|---|
| 0 | $(\mathbf{c}^0, s_c^0, s_r^0)$ | $(\mathbf{c}^0, \tilde{s}_c^0, \tilde{s}_r^0)$ | $(\mathbf{c}^0, \tilde{s}_c'^0, \tilde{s}_r'^0)$ | only content |
| 1 | $(\mathbf{c}^1, s_c^1, s_r^1)$ | $(\mathbf{c}^1, \tilde{s}_c^1, \tilde{s}_r^1)$ | $(\mathbf{c}^1, \tilde{s}_c^1, \tilde{s}_r'^1)$ | content & color |
| 2 | $(\mathbf{c}^2, s_c^2, s_r^2)$ | $(\mathbf{c}^2, \tilde{s}_c^2, \tilde{s}_r^2)$ | $(\mathbf{c}^2, \tilde{s}_c'^2, \tilde{s}_r^2)$ | content & rotation |

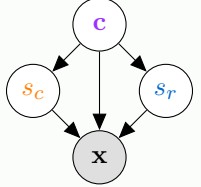

### 4.4 Causal interpretation

The described augmentation procedure can also be interpreted in causal terms (Ilse et al., 2021; Mitrovic et al., 2021; von Kügelgen et al., 2021). Given a factual observation $\mathbf{x}$, the augmented views $(\tilde{\mathbf{x}}^m, \tilde{\mathbf{x}}'^m)$ constitute pairs of *counterfactuals under joint interventions on all style variables*, provided that (i) $\mathbf{c}$ is a root note in the causal graph, to ensure content invariance in (4.3); and (ii) the style components $s_m$ do not causally influence each other, to justify the factorization in (4.6) and (4.7).[3] A causal graph compatible with these constraints is shown for Example 1.1 above on the right. As a structural causal model (SCM; Pearl, 2009), this can be written as

$$\mathbf{c} := \mathbf{u}_c, \qquad s_m := f_m(\mathbf{c}, u_m), \qquad \text{for} \qquad m = 1, \dots, M, \tag{4.9}$$

with jointly independent exogenous variables $\mathbf{u}_c$, $\{u_m\}_{m=0}^{M}$. The style conditionals $p_{\tilde{\mathbf{s}}_m|\tilde{\mathbf{s}}_m}$ in (4.6) can then arise, e.g., from *shift*, $do(s_m = f_m(\mathbf{c}, u_m) + \tilde{u}_m)$, or *perfect*, $do(\mathbf{s}_m = \tilde{u}_m)$, interventions with independent augmentation noise $\tilde{u}_m$. Note that the latter renders $\tilde{s}_m$ independent of all other variables.

### 4.5 Style identifiability and disentanglement

By construction, $\{\mathbf{c}, \tilde{s}_m\}$ is shared across $(\tilde{\mathbf{x}}^m, \tilde{\mathbf{x}}'^m)$ and can thus be identified *up to nonlinear mixing* by contrastive SSL on the $m^{\text{th}}$ dataset (von Kügelgen et al., 2021, Thm. 4.4). However, it remains unclear how to disentangle the two and recover only $\tilde{s}_m$, i.e., how to "remove" $\mathbf{c}$, which can separately be recovered as the only shared latent for $m = 0$. The following result, shows that our approach from Section 3 with $M + 1$ alignment terms and joint entropy regularization indeed disentangles and recovers the individual style components.

**Theorem 4.2** (Identifiability). *For the data generating process in* (4.1)*,* (4.7)*, and* (4.8)*, assume that*

$A_1$. *$\mathcal{Z}$ is open and simply connected; $f$ is diffeomorphic onto its image; $p_{\mathbf{z}}$ is smooth and fully supported on $\mathcal{Z}$; each $p_{\tilde{s}_m|s_m}$ is smooth and supported on an open, non-empty set around any $s_m$; the content dimensionality $d_0$ is known;*

$A_2$. *$p_{\mathbf{z}}$ and $\{p_{\tilde{s}_m|s_m}\}_{m=1}^{M}$ are such that $\{\mathbf{c}\} \cup \{\tilde{s}_m\}_{m=1}^{M}$ are jointly independent;*

$A_3$. *$d_m = 1$ for $m = 1, \dots, M$ and $\{\phi_m : \mathcal{X} \to (0,1)^{d_m}\}_{m=0}^{M}$ are smooth minimizers of*

$$\sum_{m=0}^{M} \mathbb{E}_{p_{\tilde{\mathbf{x}}, \tilde{\mathbf{x}}'}^{(m)}} \left[\left\|\phi_m(\tilde{\mathbf{x}}) - \phi_m(\tilde{\mathbf{x}}')\right\|_2\right] - H_{p_{\tilde{\mathbf{x}}}^{(0)}} \left([\phi_0(\tilde{\mathbf{x}}), \dots, \phi_M(\tilde{\mathbf{x}})]\right). \tag{4.10}$$

*Then $\phi_0$ block-identifies (von Kügelgen et al., 2021, Defn. 4.1) the content $\mathbf{c}$, and $\{\phi_m\}_{m=1}^{M}$ identify and disentangle the style latents $s_m$ in the sense that for all $m = 1, \dots, M$: $\hat{s}_m = \phi_m(\mathbf{x}) = \psi_m(s_m)$ for some invertible $\psi_m$.*

*Proof.* Part of the proof follows a similar argument as that of von Kügelgen et al. (2021, Thm. 4.4), generalized to our setting with $M + 1$ alignment terms instead of a single one, and with *joint* entropy regularization.

**Step 1.** First, we show the existence of a solution $\{\phi_m^*\}_{m=0}^{M}$ attaining the global minimum of zero of the objective in (4.10). To this end, we construct each $\phi_m^*$ by composing the inverse of the true mixing function with the cumulative distribution function (CDF) transform[4] to map each latent block to a uniform version of itself. Specifically, let $\phi_0^* := F_{\mathbf{c}} \circ f_{1:d_0}^{-1}$, and for $m = 1, \dots, M$, let $\phi_m^* := F_{\mathbf{s}_m} \circ f_{a_m:b_m}^{-1}$ with $a_m = 1 + \sum_{l=0}^{m-1} d_l$ and $b_m = \sum_{l=0}^{m} d_l$, where $F_v$ denotes the CDF of $v$. By construction, $\phi_0^*(\tilde{\mathbf{x}})$ is a function of $\mathbf{c}$ only, and uniformly distributed on $(0,1)^{d_0}$; similarly, $\phi_m^*(\tilde{\mathbf{x}})$ is a function of $\tilde{s}_m$ only and uniform on $(0,1)^{d_m}$ for $m = 1, \dots, M$. Recall that, with probability one, $\mathbf{c}$ is shared across $(\tilde{\mathbf{x}}, \tilde{\mathbf{x}}') \sim p_{\tilde{\mathbf{x}}, \tilde{\mathbf{x}}'}^{(0)}$ and $\tilde{s}_m$ is shared across $(\tilde{\mathbf{x}}, \tilde{\mathbf{x}}') \sim p_{\tilde{\mathbf{x}}, \tilde{\mathbf{x}}'}^{(m)}$. Hence, all the alignment (expectation) terms in (4.10) are zero. Finally, since $\{\mathbf{c}\} \cup \{\tilde{s}_m\}_{m=1}^{M}$ are mutually independent by assumption $A_2$, and since each $\phi_m^*$ for $m = 0, \dots, M$ is uniform on $(0,1)^{d_m}$, it follows that $[\phi_0^*(\tilde{\mathbf{x}}), \dots, \phi_M^*(\tilde{\mathbf{x}})]$ is jointly uniform on $(0,1)^{d}$. Hence, the entropy term in (4.10) is also zero.

---

[3]The allowed structure is similar to that considered by Suter et al. (2019, Fig. 1) and Wang and Jordan (2021, Fig. 9). However, ours is more general as content does not only confound different $s_m$, but also directly influences the observed $\mathbf{x}$.

[4]a.k.a. "Darmois construction" (Darmois, 1951; Hyvärinen and Pajunen, 1999; Gresele et al., 2021; Papamakarios et al., 2021).

**Step 2.** Next, let $\{\phi_m\}_{m=0}^M$ be any other solution attaining the global minimum of (4.10). By the above existence argument, this implies that (i) $\phi_m(\tilde{\mathbf{x}}) = \phi_m(\tilde{\mathbf{x}}')$ almost surely w.r.t. $p_{\tilde{\mathbf{x}},\tilde{\mathbf{x}}'}^{(m)}$ for $m = 0, \ldots, M$; and (ii) $[\phi_0(\tilde{\mathbf{x}}), \ldots, \phi_M(\tilde{\mathbf{x}})]$ is jointly uniform on $(0,1)^d$. As shown by (von Kügelgen et al., 2021, Thm. 4.4), the invariance constraint (i) together with the postulated data generating process and assumption $A_1$ implies that each $\phi_m \circ f$ can only be a function of the latents that are shared almost surely across $(\tilde{\mathbf{x}}, \tilde{\mathbf{x}}') \sim p_{\tilde{\mathbf{x}},\tilde{\mathbf{x}}'}^{(m)}$. That is, $\phi_0(\mathbf{x}) = \psi_0(\mathbf{c})$ and $\phi_m(\mathbf{x}) = \psi_m(\mathbf{c}, s_m)$ for $m = 1, \ldots, M$. By $A_1$ and constraint (ii), $\psi_0$ maps a regular density to another regular density and thus must be invertible (Zimmermann et al., 2021, Prop. 5).

**Step 3.** It remains to show that $\psi_m$ is invertible and actually cannot functionally depend on $\mathbf{c}$ for $m = 1, ..., M$, for this would otherwise violate the maximum entropy (uniformity) constraint (ii). Fix any $k \in \{1, \ldots, M\}$. By constraint (ii), $[\phi_0(\tilde{\mathbf{x}}), \phi_k(\tilde{\mathbf{x}})] = [\psi_0(\mathbf{c}), \psi_k(\mathbf{c}, \tilde{s}_k)]$ is jointly uniform on $(0,1)^{d_0+d_k}$. Hence, $\psi_0(\mathbf{c})$ and $\psi_k(\mathbf{c}, \tilde{s}_k)$ are independent. Since $\psi_0$ is invertible, this implies that $\mathbf{c}$ and $\psi_k(\mathbf{c}, \tilde{s}_k)$ are independent, too. Together with the independence of $\mathbf{c}$ and $\tilde{s}_k$ ($A_2$), it then follows from Lemma 2 of Brehmer et al. (2022) that $\psi_k$ must be constant in (i.e., cannot functionally depend on) $\mathbf{c}$. Since $k$ was arbitrary, it follows that $\phi_m(\tilde{\mathbf{x}}) = \psi_m(\mathbf{c}, \tilde{s}_m) = \psi_m(\tilde{s}_m)$ for $m = 1, \ldots, M$. Finally, invertibility of $\psi_m(\tilde{s}_m)$ for $m = 1, \ldots, M$ follows from $A_1$ and Prop. 5 of Zimmermann et al. (2021). Together with $\phi_0(\tilde{\mathbf{x}}) = \psi_0(\mathbf{c})$ (established above), this concludes the proof. $\qquad\square$

**Discussion of Thm. 4.2.** The technical assumption $A_1$ is also needed to prove content identifiability (von Kügelgen et al., 2021). Assumption $A_2$, which requires that the augmentation process renders $\mathbf{c}$ and $\{\tilde{s}_m\}_{m=1}^M$ independent, is specific to our extended analysis. It holds, e.g., if (a) $p_{\mathbf{z}}$ is such that $\mathbf{c}$ and $\{s_m\}_{m=1}^M$ are independent to begin with; or if (b) $p_{\tilde{s}|s} = p_{\tilde{s}}$ does not depend on $\mathbf{s}$, as would be the case for *perfect* interventions. As discussed in more detail in Section 6, (a) relates to work on multi-view latent correlation maximization (Lyu et al., 2021), nonlinear ICA (Gresele et al., 2019), and disentanglement (Locatello et al., 2020; Ahuja et al., 2022), whereas (b) relates to work in weakly supervised causal representation learning (Brehmer et al., 2022). In case (b), we could actually also allow for causal relations among individual style components $s_m \to s_{m'}$, as such links are broken by perfect interventions. When $A_2$ does not hold (e.g., for content-dependent style and *imperfect* interventions—arguably the most realistic setting), identifiability of the style components seems infeasible, consistent with existing negative results (Brehmer et al., 2022; Squires et al., 2023). We hypothesize that, in this case, the exogenous style variables $u_m$ in (4.9), which capture any style information not due to $\mathbf{c}$ and are jointly independent by assumption, are recovered in place of $s_m$. The assumption that $d_m = 1$ in $A_3$ is required in Step 3 of the proof to invoke Lemma 2 of Brehmer et al. (2022), which links *statistical* to *functional* independence. Whereas all other proof steps would carry through for arbitrary known style dimensions, the Lemma generally does not hold if the second argument is of greater dimensionality. It is therefore unclear how to generalize full disentanglement of all style components based on alignment and joint entropy regularization to $d_m > 1$.

## 5 Experiments

We now present our experimental results. First, we demonstrate how our approach can *completely* disentangle content from style using a simplified single-embedding-space model and synthetic datasets (Sections 5.1 and 5.2). Next, we move to a multi-embedding-space setup and the ImageNet dataset, demonstrating (i) how our approach improves downstream performance with the post-projector embedding $\mathbf{z}$ by keeping more style, and (ii) that a large performance disparity remains between $\mathbf{z}$ and the pre-projector representation $\mathbf{h}$, raising questions about the role of the projector in SSL—particularly in regard to style retention (Section 5.3).

### 5.1 Numerical dataset: Recovering only content despite varying embedding sizes

We start with a single embedding space and a simplified goal: to recover *only content* i.e., to *completely* disentangle content from style. In particular, we show how this can be achieved by appropriately tuning/adapting $\lambda_0$ in Eq. (3.2). Note that the additional style terms of Eq. (3.2) disappear in this simplified, content-only setting.

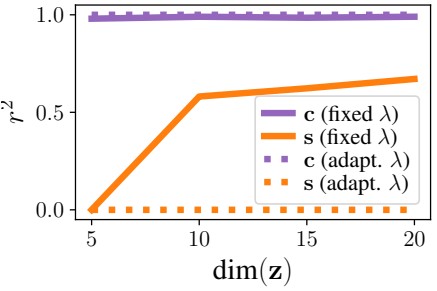

| $\lambda$ | **Augm. Strength** | **Content ($\mathbf{c}$)** | **Style ($\mathbf{s}$)** |
|---|---|---|---|
| standard | weak | 1.0 | 0.75 |
| | medium | 1.0 | 0.74 |
| | strong | 1.0 | 0.38 |
| adapted | weak | 1.0 | 0.08 |
| | medium | 1.0 | 0.05 |
| | strong | 1.0 | 0.04 |

Figure 2: **Numerical dataset: Recovering only content despite varying embedding sizes.** $r^2$ in predicting the ground-truth content $\mathbf{c}$ and style $\mathbf{s}$ from the learned embedding $\mathbf{z}$. For a fixed value of $\lambda$, excess dimensions of $\mathbf{z}$ are used to capture style. We can prevent this by adapting/increasing $\lambda$. Note: $\dim(\mathbf{c}) = 5$.

Table 2: **ColorDSprites: Recovering only content despite varying augmentation strengths.** $r^2$ in predicting ground-truth factor values from an embedding $\mathbf{z}$ learned with SimCLR. Stronger augmentations lead to more discarding of style. Adapting/increasing $\lambda$ (from its standard value) reliably discards all style, regardless of the augmentation strength. Full results in Table 5 of Appendix C.1.

### 5.1.1 Setup

**Data.** Following von Kügelgen et al. (2021, Sec. 5.1), we generate synthetic data pairs $(\mathbf{x}, \tilde{\mathbf{x}}) = (f(\mathbf{c}, \mathbf{s}), f(\mathbf{c}, \tilde{\mathbf{s}}))$ with shared content $\mathbf{c} \sim \mathcal{N}(0, \Sigma_c)$, style $\mathbf{s}|\mathbf{c} \sim \mathcal{N}(\mathbf{a} + \mathbf{Bc}, \Sigma_s)$, and perturbed style $\tilde{\mathbf{s}} \sim \mathcal{N}(\mathbf{s}, \Sigma_{\tilde{s}})$. We choose the simplest setup with $\Sigma_c$, $\Sigma_s$ and $\Sigma_{\tilde{s}}$ set to the identity, and $\dim(\mathbf{c}) = \dim(\mathbf{s}) = 5$. See von Kügelgen et al. (2021, App. D) for further details on the data-generation process.

**Training and evaluation.** We train a simple encoder $\phi$ (3-layer MLP) with SimCLR using: (i) a fixed setting $\lambda_0 = 1$, corresponding to the standard SimCLR objective; and (ii) an adaptive setting where we tune/increase $\lambda_0$ until the invariance term is sufficiently close to zero (see Appendix A.3 for details). We then report the $r^2$ coefficient of determination in predicting the ground-truth content $\mathbf{c}$ and style $\mathbf{s}$ from the learned embedding $\mathbf{z} = \phi(\mathbf{x})$.

### 5.1.2 Results

With a fixed $\lambda_0 = 1$, Fig. 2 shows how increasing the size of the learned embedding $\mathbf{z}$ makes us better able to predict the style $\mathbf{s}$. This implies that, as the embedding size $\dim(\mathbf{z})$ increases beyond that of the true content size $\dim(\mathbf{c}) = 5$, excess capacity in $\mathbf{z}$ is used to encode style (since that increases entropy). This was also observed in von Kügelgen et al. (2021, Fig. 10).

However, we can prevent this by adapting/increasing $\lambda_0$ until the invariance term is sufficiently close to zero (see Appendix A.3 for details), allowing us to recover *only content* in $\mathbf{z}$ even when there is excess capacity.

## 5.2 ColorDSprites: Recovering only content despite varying augmentation strengths

We continue with a single embedding space and the simplified goal of recovering *only content* i.e., *completely* disentangling content from style. We again show how this can be achieved by appropriately tuning/adapting $\lambda_0$ in Eq. (3.2) (where additional style terms disappear in the content-only setting), but this time with: (1) an image dataset; and (2) varying augmentation strengths.

### 5.2.1 Setup

**Data.** We use a colored version of the DSprites dataset (Locatello et al., 2019) which contains images of 2D shapes generated from 6 independent ground-truth factors (# values): color (10), shape (3), scale (6), orientation (40), x-position (32) and y-position (32). In addition, we use weak, medium and strong augmentation strengths, with examples shown in Fig. 4 of appendix A.1.

**Training and evaluation.** We train SimCLR models using: (i) a fixed setting $\lambda_0 = 1$, corresponding to the standard SimCLR objective; and (ii) an adaptive setting where we tune/increase $\lambda_0$ until the invariance term is sufficiently close to zero (see Appendix A.3 for details). We then report the $r^2$ coefficient of determination in predicting the ground-truth factor values from the learned embedding $\mathbf{z} = \phi(\mathbf{x})$ with a linear classifier.

### 5.2.2 Results

Table 2 shows that: (i) for fixed $\lambda_0$, the augmentation strengths severely affect the amount of style information in $\mathbf{z}$; and (ii) by increasing/adapting $\lambda_0$, we can reliably remove (almost) all style, regardless of the augmentation strengths. Table 5 of Appendix C.1 gives the full set of results, including per-factor $r^2$ values and the corresponding results for VICReg.

### 5.3 ImageNet: Improving downstream performance by keeping more style

We now move to multiple embedding spaces and a more realistic dataset, with the goal of ==keeping more style information in the embedding space==.

### 5.3.1 Setup

**Data.** We use a blurred-face ImageNet1k (Russakovsky et al., 2015) dataset and the standard augmentations or transformations (random crop, horizontal flip, color jitter, grayscale and blur). See Appendix A.2 for further details.

**Training.** We train all models for 100 epochs and use both SimCLR (Chen et al., 2020a) and VICReg (Bardes et al., 2022) as the base methods. For our method (+ Ours), we group the transformations into spatial (crop, flip) and appearance (color jitter, greyscale, blur) transformations, giving rise to $M = 2$ style embedding-spaces. We also train two versions of our method: *Ours-FT*, which fine-tunes a base model, *Ours-Scratch*, which trains a model from scratch (random initialization) with our multi-embedding-space objective (i.e., Eq. (3.2)). For VICReg, we found training with our method from scratch (*VICReg + Ours-Scratch*) to be quite unstable, so this is excluded. See Appendix A.2 for further implementation details.

**Evaluation.** We follow the setup of Ericsson et al. (2021) to evaluate models on 13 diverse downstream tasks, covering object/texture/scene classification, localization, and keypoint estimation. We then group these datasets/tasks by the information they most depend on, Spatial, Appearance or Other (not spatial- or appearance-dominant), resulting in the following groups:

- *Spatial:* The Caltech Birds (CUB) (Wah et al., 2011) tasks of bounding-box prediction ($\text{CUB}_{bbox}$) and keypoint prediction ($\text{CUB}_{kpts}$).

- *Appearance:* The Describable Textures Dataset (DTD, Cimpoi et al. 2014), and the color-reliant tasks of Oxford Flowers (Nilsback and Zisserman, 2008) and Oxford-IIIT Pets (Parkhi et al., 2012).

- *Other:* FGVC Aircraft (Maji et al., 2013), Caltech-101 (Li et al., 2022), Stanford Cars (Krause et al., 2013), CIFAR10 (Krizhevsky, 2009), CIFAR100 (Krizhevsky, 2009), Caltech Birds (CUB, Wah et al. 2011) class prediction ($\text{CUB}_{cls}$), SUN397 (Xiao et al., 2010) and VOC2007 (Everingham et al., 2007).

### 5.3.2 Results

**Post-projector embedding z.** Fig. 3 (left) shows that: (i) using our method to fine-tune (+ *Ours-FT*) boosts downstream performance for both SimCLR and VICReg variants, implying that more style information has been kept post-projector; and (ii) using our method to train a model from scratch (+ *Ours-Scratch*) leads to an even bigger boost in downstream performance, i.e., even more style retention post projector. Furthermore, for *SimCLR + Ours-Scratch*, note the *improved downstream performance despite lower ImageNet performance*—the pre-training task for which the transformations were designed. As discussed in Example 1.1, the discarding of style information tends to help pre-training performance but hurt downstream performance, motivating our style-retaining algorithms.

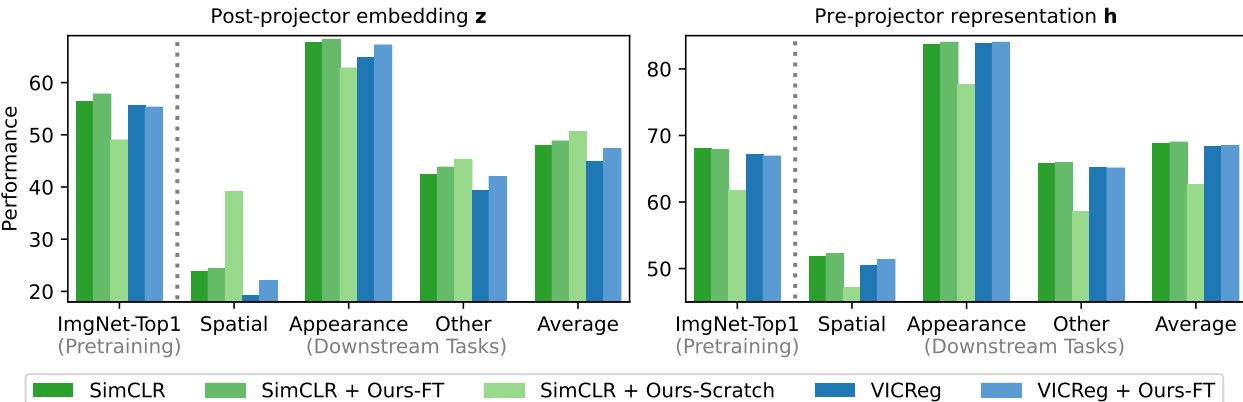

Figure 3: **ImageNet: Improving downstream performance by keeping more style.** We report linear-probe performance on ImageNet and *grouped* downstream tasks when using both **z** (left, post projector) and **h** (right, pre projector). 13 downstream tasks are grouped by the information they most depend on: spatial, appearance or other (not spatial- or appearance-dominant). We also report the average over tasks (see Table 7 of Appendix C.2 for per-task results) rather than over groups. All bars show top-1 accuracy (%) except for those in the Spatial group, which show $r^2$.

**Pre-projector representation h.** Unfortunately, this improved performance with the post-projector **z** did not translate into improved performance with the pre-projector **h**. In particular, Fig. 3 (right) shows that: (i) training with our method from scratch hurts downstream performance when using **h**, despite improved performance when using **z**; (ii) fine-tuning with our method only leads to very marginal gains when using **h**, despite improved performance when using **z**; and (iii) downstream performance is still significantly better with **h** than **z**, as shown by the gap of 20 percentage points.

**Summary.** These results validate the ability of our method to keep more style information post-projector but raise even more questions about the role of the projector in self-supervised learning (Bordes et al., 2023; Jing et al., 2022; Gupta et al., 2022; Xue et al., 2024).

# 6 Related Work

**Self-supervised learning.** Prior works in self-supervised learning have investigated different ways to retain more style information, including the prediction of augmentation parameters (Lee et al., 2020; 2021), seeking transformation equivariance (Dangovski et al., 2022), employing techniques that improve performance when using a linear projector (Jing et al., 2022), and, most related to our approach, learning multiple embedding spaces which seek different invariances (Xiao et al., 2021). Table 3 presents the key differences between our framework and that of Xiao et al. (2021). While both frameworks learn multiple embedding spaces using structure augmentations, only ours: 1) learns *disentangled* embedding spaces, due to the *joint entropy* term in Eq. (3.3) ensuring no duplicate or redundant information; and 2) provides a *theoretical analysis* with the conditions under which the framework recovers/identifies the underlying style attributes of the data. Additionally, at the implementation level, our construction of image views permits more negative samples for a given batch size, as depicted in Fig. 5 of Appendix B.

**Disentanglement in generative models.** In a generative setting, disentangled representations are commonly sought (Desjardins et al., 2012; Higgins et al., 2017; Eastwood and Williams, 2018; Eastwood et al., 2023). In the vision-as-inverse-graphics paradigm, separating or disentangling "content" and "style" has a long history (Tenenbaum and Freeman, 1996; Kulkarni et al., 2015). Purely based on i.i.d. data and without assumptions on the model-class, disentangled representation learning is generally impossible (Hyvärinen and Pajunen, 1999; Locatello et al., 2019). More recently, generative disentanglement has thus been pursued with additional weak supervision in the form of paired data (Bouchacourt et al., 2018; Locatello et al., 2020).

Table 3: **High-level comparison with Xiao et al. (2021).** While both use structured augmentations and multiple embedding spaces to capture style attributes of the data, only ours seeks disentangled embedding spaces and provides theoretical grounding/analysis.

| Method | Structured augmentations | Multiple embeddings | Disentangled embeddings | Theoretical underpinning |
|---|---|---|---|---|
| Xiao et al. (2021) | ✔ | ✔ | ✘ | ✘ |
| Ours | ✔ | ✔ | ✔ | ✔ |

**Identifiability in disentangled and causal representation learning.** Our Thm. 4.2 can be viewed as an extension of the content block-identifiability result of von Kügelgen et al. (2021, Thm. 4.4), which was generalized to a multi-modal setting with distinct mixing functions $f_1 \neq f_2$ and additional modality-specific latents by Daunhawer et al. (2023, Thm. 1). The two options discussed at the end of Section 4 for satisfying assumption $A_2$—(a) independent style variables, and (b) perfect interventions—can be used to draw additional links to existing identifiability results. Option (a) relates to a result of Lyu et al. (2021, Thm. 2) showing that $\tilde{\mathbf{s}}$ and $\tilde{\mathbf{s}}'$ can be block-identified through latent correlation maximization with invertible encoders, provided that $\mathbf{c}$, $\tilde{\mathbf{s}}$, and $\tilde{\mathbf{s}}'$ are mutually independent. Note, however, that Thm. 4.2 establishes a more fine-grained disentanglement into individual style components. On the other hand, Gresele et al. (2019) and Locatello et al. (2020) prove identifiability of individual latents for the setting in which all latents are mutually independent and subject to change (with probability $> 0$), i.e., without an invariant block of content latents. Option (b) relates to a result of Brehmer et al. (2022, Thm. 1) showing that all variables (and the graph) in a causal representation learning setup can be identified through weak supervision in the form of pairs $(\mathbf{x}, \tilde{\mathbf{x}})$ arising from single-node perfect interventions by fitting a generative model via maximum likelihood.

Closely related is also the work of Ahuja et al. (2022) who do not assume independence of latents, and also consider learning from $M$ views arising from sparse perturbations, but require perturbations on all latent blocks for full identifiability. The concurrent work of Yao et al. (2024) also studies the learning of identifiable representations from two or more views via contrastive SSL. They focus on block-identifiability in a partially-observed setting, where each view only depends on a subset of latent variables, and take a multi-modal perspective (rather than the counterfactual, augmentation-based perspective explored in this work).

## 7 Conclusion

We have presented a self-supervised framework that uses structured data augmentations to learn disentangled representations. By formalizing this framework from a causal-latent-variable-model perspective, we proved that it can recover not only invariant content latents, but also disentangle individual varying style latents. Experimentally, we demonstrated how our framework can fully disentangle content from style and how, by retaining more style information, it improves downstream performance with the post-projector embedding.

Future work may investigate new types of data augmentations that are designed for disentangling rather than discarding, as well as their use for self-supervised (causal) representation learning. Future work may also investigate the role of the projector in style retention, as well as its interplay with explicit style-retaining objectives.

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

# Appendices

## A  Implementation Details

**Embedding dimensionality.** For fair comparison, we fix the dimension of the embedding for all methods—2048 for ImageNet and 256 for ColorDSprites. For our method, we split the embedding into $M + 1$ pieces according to pre-specified proportions. More specifically, for ImageNet, where we use $M = 2$ style embedding spaces (spatial and appearance groups), the fractions were 0.5 for content, 0.25 for spatial and 0.25 for appearance.

### A.1  ColorDSprites

Fig. 4 depicts samples from the ColorDprites dataset when transformed with transformations/augmentations of different strengths.

Table 4: **Augmentation parameters for weak, medium and strong transformations on ColorD-Sprites.** Scale factors are applied using the PyTorch functions `Pad` (zoom out) and `CentreCrop` (zoom in) with a sampling factor $x_s \sim \mathrm{Bern}([z_{\mathrm{out}}, z_{\mathrm{in}}], [0.5, 0.5])$, where $z_{\mathrm{out}} \sim U(0, 0.1(s-1))$, $z_{\mathrm{in}} \sim U(1, s)$. Rotation is sampled as degrees $r \sim U(-X, X)$, and translation uses a maximum fraction for horizontal and vertical adjustments, $t_x \sim U(-\mathrm{image\_width} \times x, \mathrm{image\_width} \times x)$.

| Parameter | Level | Values |
|---|---|---|
| Scale | weak | 1.5 |
| | medium | 2.0 |
| | strong | 3.0 |
| Color | weak | {brightness: 0.2, saturation: 0.2, contrast: 0.2, hue: 0.1, p: 0.8, gray_p: 0.0} |
| | medium | {brightness: 0.4, saturation: 0.4, contrast: 0.4, hue: 0.3, p: 1.0, gray_p: 0.2} |
| | strong | {brightness: 0.8, saturation: 0.8, contrast: 0.8, hue: 0.5, p: 1.0, gray_p: 0.2} |
| Rotation | weak | {p: 0.5, degrees: 20} |
| | medium | {p: 0.5, degrees: 60} |
| | strong | {p: 0.5, degrees: 180} |
| Translation | weak | {x: 0.15, y: 0.15} |
| | medium | {x: 0.2, y: 0.2} |
| | strong | {x: 0.25, y: 0.25} |

### A.2  ImageNet

**Pretraining.** Our ImageNet1k pretraining setup is based on the settings in Bardes et al. (2022), which can be consulted for full details. We train ResNet50 (He et al., 2016) models for only 100 epochs, with 3-layer projectors of dimension 8196. The optimizer is LARS (You et al., 2017; Goyal et al., 2017), the batch size is 2048 and the learning rate follows a cosine decay schedule (Loshchilov and Hutter, 2017).

The data augmentation also follows Bardes et al. (2022) and is applied asymetrically to the two views. It includes crops, flips, color jitter, grayscale, solarize and blur. These atomic augmentations are split into two groups: *spatial* (crops and flips) and *appearance* (color jitter, grayscale, solarize and blur). Thus, the number of "style" attributes in this setting are $M = 2$.

While we aim for fair experiments that use default hyperparameters, projectors, and augmentation settings, we note that these are optimized for existing SSL methods that prioritize information removal. Perhaps other settings, such as different augmentations explored in Xiao et al. (2021) and Lee et al. (2021), can be beneficial in our framework which instead aims to retain and disentangle information.

**Downstream evaluation.** Our downstream evaluation follows that of Ericsson et al. (2021). We train linear models (logistic or ridge regression) on frozen pre-projector representations $\mathbf{h}$ *and* post-projector embeddings $\mathbf{z}$. Images are cropped to $224 \times 224$, with $L2$ regularization searched using 5-fold cross-validation over 45 logarithmically spaced values in the range $10^{-6}$ to $10^{5}$.

### A.3  Selecting $\lambda_m$

We now describe our procedure for choosing our $\lambda_m$ hyperparameters in Eq. (3.2). As a reminder, our goal is to set $\lambda_m$ such that the invariance terms are sufficiently close to zero, since we want each embedding space *to fully achieve its particular invariance*. Note how this differs from the standard goal in self-supervised learning, where one usually chooses an invariance-entropy trade-off (see Eq. (2.1) and Table 1) that maximizes performance on the pretraining task (e.g., ImageNet Top-1 accuracy). For this reason, we place the $\lambda_m$'s on invariance terms $\mathcal{L}^{\text{inv}}$ rather than the entropy term $\mathcal{L}^{\text{ent}}$, as done in some prior works (Wang and Isola, 2020; Zbontar et al., 2021; Bardes et al., 2022).

While we could choose our $\lambda_m$'s using a fine-grained grid search, such that this invariance term is close to zero at the end of training, we instead choose to iteratively adapt the $\lambda_m$'s during training using a dual-ascent approach. In particular, given a step size or learning rate $\eta$ and tolerance level $\epsilon$, we perform iterative gradient-based updates of $\lambda_m$ during training using

$$\lambda_m^t \leftarrow \lambda_m^{t-1} + \eta \cdot \text{relu}(\mathcal{L}^{\text{inv}}(\theta^t) - \epsilon), \tag{A.1}$$

where the model parameters $\theta^t$ at step $t$ are updated during an "inner loop", and the hyperparameters $\lambda_m^t$'s at step $t$ are updated during the "outer loop".

## B  Visual comparison with Xiao et al. (2021)

Fig. 5 shows how our construction of image views allows more negative samples in a given batch, compared to the query-key construction of Xiao et al. (2021).

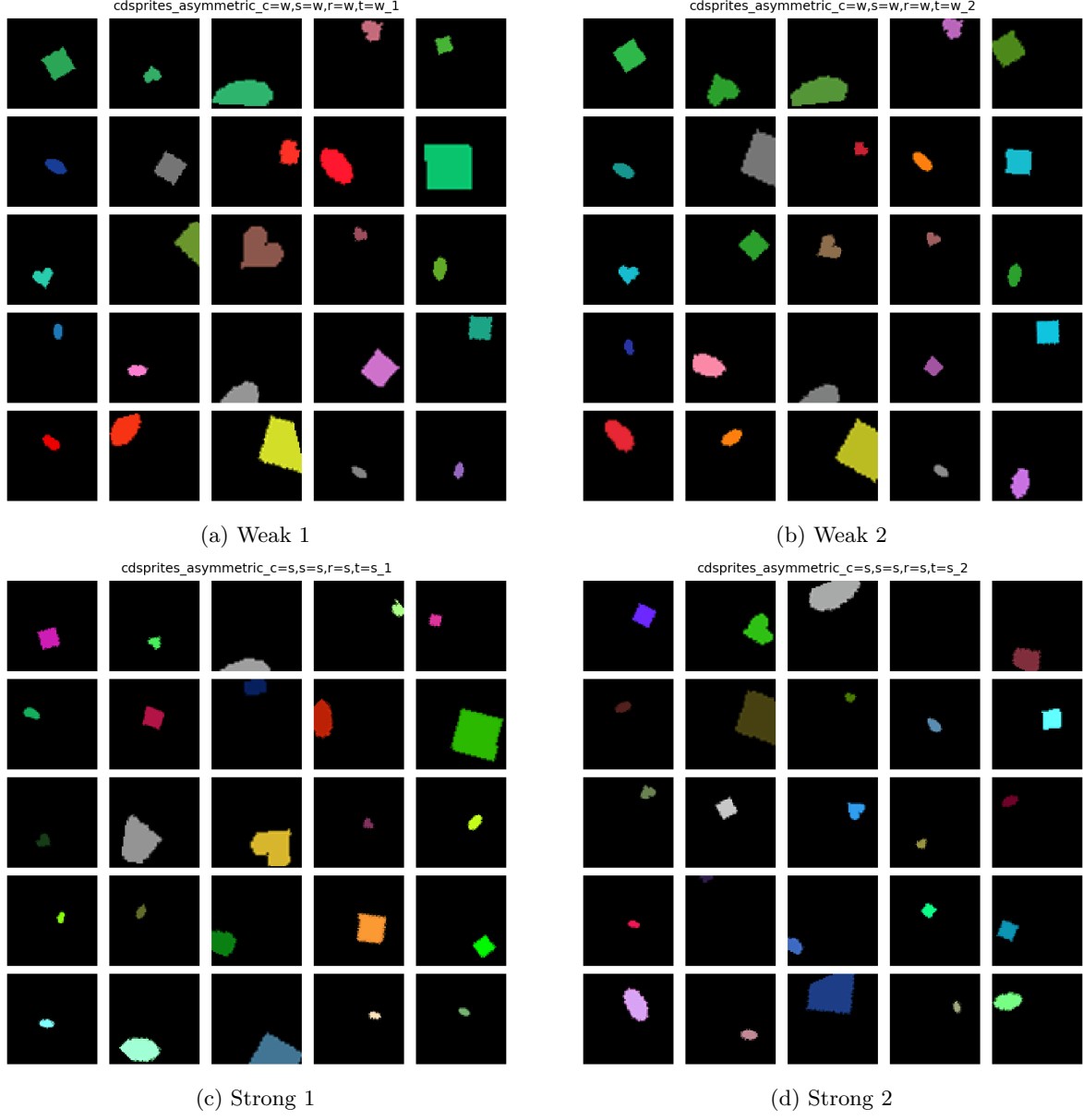

Figure 4: **ColorDSprites: Varying augmentation strengths.** Columns show augmentation pairs of the same strength. Note that images are more similar across (a) & (b) than across (c) & (d), in terms of the following style attributes: color, orientation, scale, translation and X-Y position.

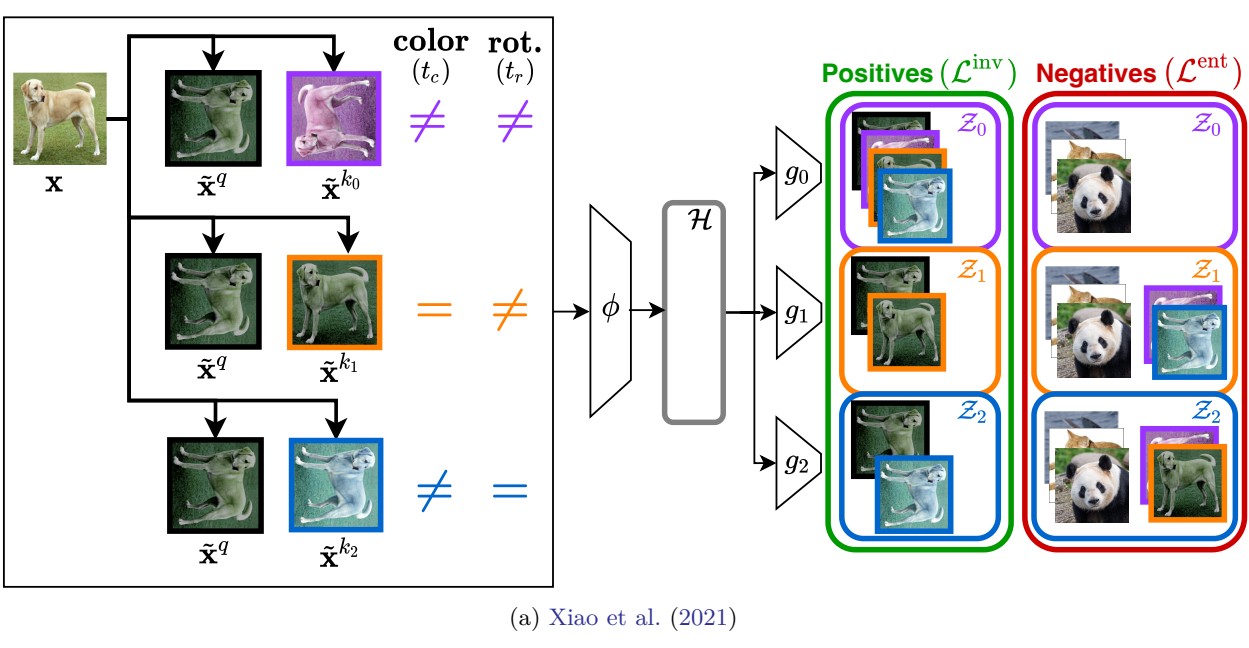

(a) Xiao et al. (2021)

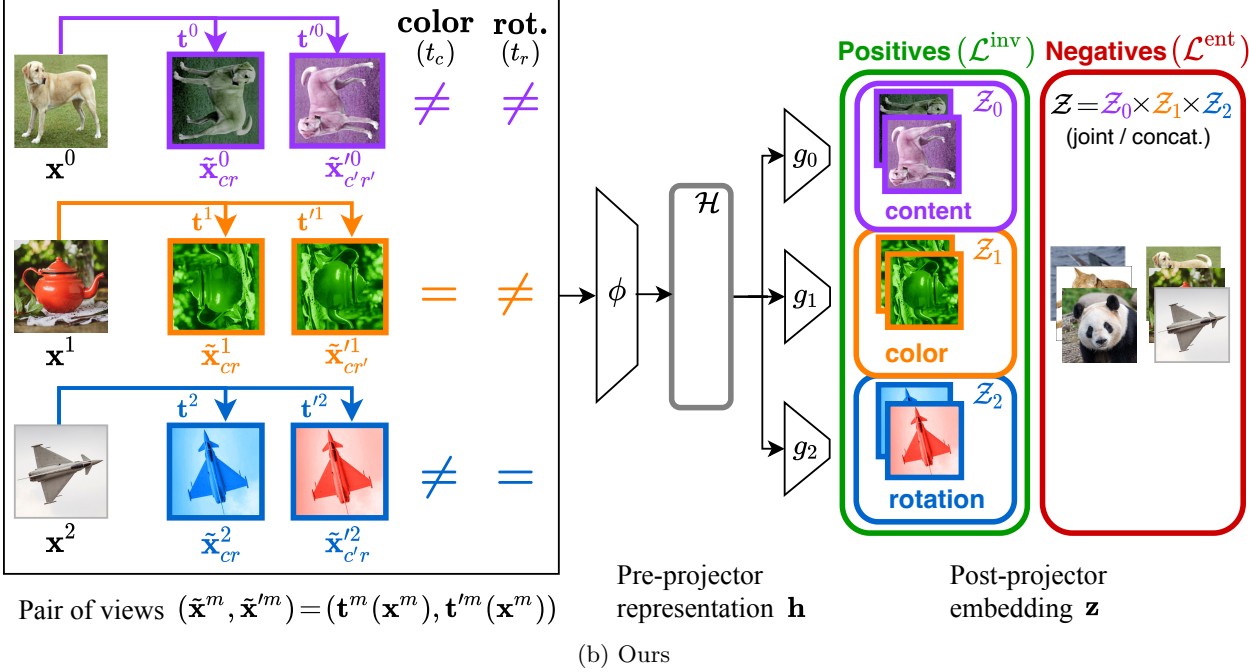

(b) Ours

Figure 5: **Comparison with Xiao et al. (2021).** Note the differences in data augmentation modules, as well as the embedding spaces in which positives and negatives are compared. In particular, note the number of different images in a given batch, with our framework containing more true negatives. See Xiao et al. (2021, Sec. 3) for details on their query-key notation.

Table 5: **ColorDSprites: Recovering only content despite varying augmentation strengths.** $r^2$ in predicting ground-truth factor values from a learned embedding **z**. Stronger augmentations lead to more discarding of style. Adapting/increasing $\lambda$ (from its standard value) reliably discards all style, regardless of the augmentation strengths.

| Algorithm | $\lambda$ | Augm. Strength | Content (c) | Style (s) | | | | | |
|---|---|---|---|---|---|---|---|---|---|
| | | | Shape | Color | Scale | Orient. | PosX | PosY | Avg. |
| SimCLR | standard | weak | 1.0 | 0.93 | 0.89 | 0.30 | 0.82 | 0.83 | 0.75 |
| | | medium | 1.0 | 0.73 | 1.00 | 0.19 | 0.89 | 0.89 | 0.74 |
| | | strong | 1.0 | 0.31 | 1.00 | 0.05 | 0.23 | 0.30 | 0.38 |
| | adapted | weak | 1.0 | 0.21 | 0.17 | 0.00 | 0.01 | 0.01 | 0.08 |
| | | medium | 1.0 | 0.10 | 0.16 | 0.00 | 0.00 | 0.00 | 0.05 |
| | | strong | 1.0 | 0.10 | 0.11 | 0.00 | 0.00 | 0.00 | 0.04 |
| VICReg | standard | weak | 1.0 | 0.87 | 0.71 | 0.29 | 0.45 | 0.45 | 0.55 |
| | | medium | 1.0 | 0.40 | 1.00 | 0.05 | 0.56 | 0.56 | 0.51 |
| | | strong | 1.0 | 0.12 | 0.99 | 0.08 | 0.62 | 0.62 | 0.49 |
| | adapted | weak | 1.0 | 0.20 | 0.17 | 0.00 | 0.00 | 0.00 | 0.07 |
| | | medium | 1.0 | 0.10 | 0.52 | 0.00 | 0.00 | 0.01 | 0.13 |
| | | strong | 1.0 | 0.10 | 0.53 | 0.00 | 0.00 | 0.00 | 0.13 |

## C   Further Results

We now present additional results.

### C.1   ColorDSprites

Table 5 gives the full, per-factor results for SimCLR and VICReg when trying to predict ground-truth content (**c**) and style (**s**) values from the post-projector embedding (**z**). As an expanded version of Table 2 in Section 5.2, Table 5 shows how different augmentation strengths affect which style attributes are recovered in the post-projector embedding **z**. It also reinforces the message from Section 5.2: adaptively increasing $\lambda_0$ ensures that we reliably remove all style from our content space.

### C.2   ImageNet

**Grouped results.**   As detailed in section 5.3, we group the downstream tasks into those which rely on spatial, appearance and other (not spatial- or appearance-heavy) information. The results for these groupings are given Table 6 and depicted visually in Fig. 3. As a reminder, the groupings are:

- **Spatial**: $CUB_{bounding\text{-}box}$, $CUB_{keypoints}$.
- **Appearance**: DTD, Flowers, Pets.
- **Other**: Aircraft, Caltech101, Cars, CIFAR10, CIFAR100, $CUB_{class}$, SUN, VOC.

**Full/ungrouped results.**   Table 7 gives the full ungrouped results, showing the performance for each individual downstream task.

Table 6: **Grouped ImageNet results.** We report linear-probe performance on ImageNet and *grouped* downstream tasks. All columns show top-1 accuracy (%) except "Spatial", which shows the $r^2$ coefficient of determination for downstream regression tasks. In the final column, the average is over tasks (shown in Table 7) rather than over groups (shown in this table).

| Algorithm | Features | Pretraining (↑) | Downstream tasks (↑) | | | |
|---|---|---|---|---|---|---|
| | | ImageNet Top-1 | Spatial | Appearance | Other | Average |
| SimCLR | z | 56.5 | 23.8 | 67.8 | 42.4 | 47.9 |
| SimCLR + Ours-FT | z | **57.8** | 24.4 | **68.3** | 43.8 | 48.8 |
| SimCLR + Ours-Scratch | z | 49.0 | **39.1** | 62.8 | **45.3** | **50.6** |
| VICReg | z | 55.7 | 19.2 | 64.8 | 39.4 | 45.0 |
| VICReg + Ours-FT | z | 55.3 | 22.0 | 67.2 | 42.0 | 47.3 |
| SimCLR | h | **68.1** | 51.9 | 83.7 | **65.9** | 68.9 |
| SimCLR + Ours-FT | h | 67.9 | **52.2** | **84.0** | **65.9** | **69.0** |
| SimCLR + Ours-Scratch | h | 61.7 | 47.1 | 77.6 | 58.5 | 62.6 |
| VICReg | h | 67.2 | 50.5 | 83.9 | 65.3 | 68.4 |
| VICReg + Ours-FT | h | 66.9 | 51.3 | **84.0** | 65.1 | 68.5 |

Table 7: **Full ImageNet results.** We report linear-probe performance on ImageNet and a broad range of downstream tasks, showing top-1 accuracy (%) for all but $CUB_{bbox}$ ($r^2$, i.e., the coefficient of determination), $CUB_{kpt}$ ($r^2$), and VOC ($AP_{50}$, i.e., mean average precision with an IoU threshold of 0.5). We use frozen embeddings **z** (post-projector) and representations **h** (pre-projector). Our method is used to fine-tune a base SimCLR/VICReg model (+ Ours-FT) or to train a model from scratch (+ Ours-Scratch). Ct101: CalTech101. Cf10: CIFAR10.

| Algorithm | Feat. | ImgNt | Acft | Ct101 | Cars | Cf10 | Cf100 | $CUB_{bbox}$ | $CUB_{cls}$ | $CUB_{kpt}$ | DTD | Flwrs | Pets | SUN | VOC | Avg. |
|---|---|---|---|---|---|---|---|---|---|---|---|---|---|---|---|---|
| SimCLR | z | 56.5 | 14.6 | 70.9 | 13.0 | 76.7 | 50.5 | 35.6 | 22.5 | 12.0 | 66.4 | 66.8 | 70.3 | 48.9 | 74.6 | 47.9 |
| SimCLR + Ours-FT | z | **57.8** | 15.9 | 72.4 | 14.6 | 77.8 | 53.6 | 36.2 | 22.5 | 12.6 | 67.0 | 67.3 | 70.6 | 49.5 | 74.9 | 48.8 |
| SimCLR + Ours-Scratch | z | 49.0 | 25.9 | 77.6 | 14.4 | 81.8 | 56.2 | 60.5 | 15.1 | 17.6 | 64.4 | 63.4 | 60.7 | 45.9 | 73.6 | **50.6** |
| VICReg | z | 55.7 | 10.6 | 69.5 | 9.5 | 75.0 | 48.3 | 27.6 | 17.1 | 10.8 | 64.6 | 61.3 | 68.4 | 46.1 | 75.6 | 45.0 |
| VICReg + Ours-FT | z | 55.3 | 11.7 | 72.6 | 11.1 | 78.5 | 54.4 | 32.5 | 17.8 | 11.4 | 66.5 | 66.8 | 68.3 | 48.0 | 75.5 | 47.3 |
| SimCLR | h | **68.1** | 50.9 | 88.2 | 50.7 | 89.3 | 73.0 | 71.3 | 48.6 | 32.5 | 75.2 | 93.5 | 82.4 | 60.3 | 79.7 | 68.9 |
| SimCLR + Ours-FT | h | 67.9 | 51.0 | 88.1 | 51.0 | 89.4 | 72.9 | 71.4 | 48.5 | 32.9 | 75.9 | 93.5 | 82.6 | 60.2 | 79.7 | **69.0** |
| SimCLR + Ours-Scratch | h | 61.7 | 46.2 | 86.5 | 37.5 | 87.2 | 67.4 | 70.6 | 29.6 | 23.6 | 72.6 | 86.8 | 73.3 | 55.1 | 77.2 | 62.6 |
| VICReg | h | 67.2 | 51.1 | 87.6 | 52.6 | 88.3 | 70.1 | 69.1 | 47.4 | 31.9 | 75.3 | 93.4 | 83.1 | 59.7 | 79.6 | 68.4 |
| VICReg + Ours-FT | h | 66.9 | 50.1 | 87.5 | 52.4 | 88.4 | 70.3 | 69.8 | 47.2 | 32.8 | 75.7 | 93.6 | 82.7 | 59.8 | 79.6 | 68.5 |

