# OpenReview forum: "Self-Supervised Disentanglement by Leveraging Structure in Data Augmentations"
_TMLR — Rejected by TMLR_

### Review · Reviewer_hV11 · 2024-10-11

**Summary Of Contributions:**

This paper presents a new approach to self-supervised representation learning (SSL). Traditional SSL (e.g. SimCLR) uses a single representation that is trained to be invariant to a bunch of specified augmentations. Here, the idea is, given a set of $M$ augmentations, to train jointly $M+1$ representations, one invariant to all specified augmentations, and each other one invariant to all-but-one augmentations. This resembles leave-one-out contrastive learning (Xiao et al., 2021) although the loss functions are not exactly the same. The authors also provide a causal perspective on their approach, and prove an identifiability theorem building on von Kügelgen et al. (2021). They show experimentally that their approach allows to learn more versatile representations than traditional SSL.

**Audience:**

Yes

**Broader Impact Concerns:**

I have no particular concerns.

**Claims And Evidence:**

Yes

**Requested Changes:**

- As mentioned above, empirical comparisons with Xiao et al. (2021) would strengthen the paper.

- One thing I found unclear in the experiments was the dimension of the representation. Is the size of the representation of your method as big as the one of the standard methods like SimCLR, or is it $M+1$ times bigger? How many additional parameters does your method bring about?

- It would be interesting to see (perhaps as an appendix), the performance of the individual embeddings $z_0, \ldots, z_M$. In particular, it would be interesting to see if $z_0$ is (as one would expect) one of the best according to the standard ImgNet-Top1 metric.

**Strengths And Weaknesses:**

**Strenghts**

This is a extremely well-written paper, with a compelling story. Both methodological and theoretical parts are very clear, and the experiments are well-motivated and sensible. In particular, I thought the experiment on Imagenet was very interesting (and appreciated the honesty of mentioning that the results of the pre-projector representation were disappointing).

While I am not an expert of self-supervised learning, I believe that the authors did a good job at citing relevant related work, in particular Xiao et al. (2021) and von Kügelgen et al. (2021), which are the closest works.

**Weaknesses**

Empirical comparisons with Xiao et al. (2021) would strengthen the paper. In particular, it would be important to have more support to the claim that the method of Xiao et al. (2021) does "not achieve disentanglement" (as claimed, e.g. at the end of Section 3).

---

> ### Author Response · Authors · 2024-10-31
> **Response to Reviewer hV11**
>
> We sincerely thank the reviewer for their time, thorough review and kind words. We address the requested changes below.
>
> > _“Empirical comparisons with Xiao et al. (2021) would strengthen the paper.”_
>
> We agree that these would strengthen the paper. However, we note that:
> - **Our claim that they do “not achieve disentanglement” is based on their learning objective rather than their empirical results.** Since “content” is invariant to all transformations (by definition) and Xiao et al. use a per-embedding-space entropy term (rather than a joint term), they get $M$ additional copies of content—one per style space. Thus, they cannot achieve disentanglement—content is tangled into _every_ style embedding space! We have added a line to the very end of Section 3 to clarify this important point.
> - **Xiao et al. (2021) do not provide code.** We did email the authors, but only received minor components. Despite our best efforts, we could not reproduce their results, and they did not report results on ImageNet1k (but rather the smaller ImageNet100).
>
> > _“One thing I found unclear in the experiments was the dimension of the representation. Is the size of the representation of your method as big as the one of the standard methods like SimCLR, or is it times bigger? How many additional parameters does your method bring about?”_
>
> This is a great question and it should be clearer in the paper. For a fair comparison, we fix the dimension of the embedding for all methods—2048 for ImageNet and 256 for ColorDSprites. For our method, we split the embedding into $M+1$ pieces according to pre-specified proportions/fractions. More specifically, for ImageNet, where we used $M=2$ style embedding spaces (spatial and appearance groups), the fractions were 0.5 for content, 0.25 for spatial and 0.25 for appearance. **We have updated the start of Appendix A to make this clear.**
>
> > _“It would be interesting to see (perhaps as an appendix), the performance of the individual embeddings $z_0, \dots, z_M$. In particular, it would be interesting to see if $z_0$ is (as one would expect) one of the best according to the standard ImgNet-Top1 metric.”_
>
> We agree that this would be an interesting experiment, and would also expect $z_0$ to be best (since the augmentations, their strengths, and other hyperparameters have been designed and optimized for ImageNet Top1 performance). Unfortunately, due to a job change of the corresponding author, we no longer have access to the models or capacity to re-run these experiments.

---

> > ### Comment · Reviewer_hV11 · 2024-11-18
> >
> > Many thanks for your clarifications and the revision!

---

### Review · Reviewer_H2K4 · 2024-10-14

**Summary Of Contributions:**

This proposal proposes to disentangle the style variables, instead of discarding them, in data augmentation for self-supervised learning.

**Audience:**

Yes

**Broader Impact Concerns:**

None.

**Claims And Evidence:**

No

**Requested Changes:**

Please address the concerns listed in the Weaknesses section.

**Strengths And Weaknesses:**

Strengths:

1. Previous analyses of SSL with data augmentations considered style latents
as nuisance variables to be discarded; in contrast, this paper seeks to
identify and disentangle different style variables.

2. Some theoretical results using causal analysis are provided.

3. Several simulation studies are shown.

Weaknesses:

1. The experiments using real-world data are very limited, which does not appear to
sufficiently support the usefulness of the proposed approach.

2. It is still not so convincing that the proposed approach would work well
in real world. This method assumes M style variables/transformations and seeks
to disengle them to retain one of them while discarding the others.
For downstream learning tasks, even if the task is known, it seems still unknown
what kind of style variables to keep and what to discard.

3. The experimental results with pre-projector representation h do not appear to have improvements
compared to baseline models.

4. The theoretical results appear to be direct/straightforward extension of
the theoretical results developed in (von Kügelgen et al., 2021), which seeks
a pure content-based representation by discarding all style variables/augmentations.
This paper considers M style variables and seeks to keep one while discarding the rest.

---

> ### Author Response · Authors · 2024-10-31
> **Response to Reviewer H2K4**
>
> We sincerely thank the reviewer for their time and review. We address the concerns below.
>
> ## Weaknesses
> > _“The experiments using real-world data are very limited, which does not appear to sufficiently support the usefulness of the proposed approach.”_
>
> First, **we disagree that the experiments on real-world data are limited (in scope)** since:
> - We report results on ImageNet—the standard for SSL methods.
> - Most style-retaining works [1,2] only report results on the smaller 100-class ImageNet100—we use ImageNet1000;
> - Most SSL works only report downstream performance on a handful (usually between one and six) of downstream tasks [1–4] — we report results on thirteen diverse downstream tasks, covering object/texture/scene classification, localization, and keypoint estimation (as detailed in Section 5.3.1).
>
> Second, we believe that **our experiments sufficiently support the effectiveness of our method in keeping more style information in the space which is optimized (i.e., the embedding space $\mathbf{z}$)—the main empirical claim of our paper**. As noted by Reviewer hV11, we tried to be extremely honest about these results and not overclaim. However, in response to the quoted comment above, **we have further toned down the empirical claims of the paper to ensure that they are fully supported by the experimental evidence** (see the general response for a list of changes). We hope that the reviewer now agrees that all empirical claims are fully supported by the experimental evidence. However, if we have missed anything, or if there are remaining concerns/suggestions, please let us know.
>
>
> > _“It is still not so convincing that the proposed approach would work well in real world. This method assumes $M$ style variables/transformations and seeks to disengle them to retain one of them while discarding the others. For downstream learning tasks, even if the task is known, it seems still unknown what kind of style variables to keep and what to discard.”_
>
> There appears to be some confusion here. **We seek to disentangle and keep _all_ $M$ style variables** (each in a separate style space, as illustrated in Fig. 1)—precisely because the downstream task is unknown (and thus we do not know which style variables to keep/discard).
>
> > _“The experimental results with pre-projector representation $h$ do not appear to have improvements compared to baseline models.”_
>
> Yes, as noted by Reviewer hV11, we tried to be extremely honest about these results and not overclaim (we state this fact in plain terms in Section 5.3.2, second paragraph). We still believe that our experiments, approach and work carry value for the community—**please see our general response for more details on this point**.
>
>
> > _“The theoretical results appear to be direct/straightforward extension of the theoretical results developed in (von Kügelgen et al., 2021), which seeks a pure content-based representation by discarding all style variables/augmentations. This paper considers $M$ style variables and seeks to keep one while discarding the rest.”_
>
> The same confusion appears here again: **we seek to disentangle and keep _all_ $M$ style variables, not just one**.
>
> We acknowledge that the result builds on the cited prior work and are open about this in the manuscript. However, **we disagree that this is a “direct/straightforward extension”**. In particular, separating and recovering all style variables requires additional assumptions ($A_2$ and $A_3$), a different objective function (Eq. (4.10)), and new proof arguments as explained in Steps 1–3.
>
>
> ## Claims And Evidence
> May we ask what claims the reviewer believes to be presented without supporting evidence? And if this is still the case in the updated manuscript?
>
>
> [1] Xiao, T., Wang, X., Efros, A. A., & Darrell, T. (2021). What Should Not Be Contrastive in Contrastive Learning. In _International Conference on Learning Representations_.
>
> [2] Lee, H., Lee, K., Lee, K., Lee, H., & Shin, J. (2021). Improving transferability of representations via augmentation-aware self-supervision. In _Advances in Neural Information Processing Systems 34_, 17710-17722.
>
> [3] Bordes, F., Balestriero, R., Garrido, Q., Bardes, A., & Vincent, P. (2024). Guillotine Regularization: Why removing layers is needed to improve generalization in Self-Supervised Learning. In _Transactions on Machine Learning Research_.
>
> [4] Jing, L., Vincent, P., LeCun, Y., & Tian, Y. (2022). Understanding Dimensional Collapse in Contrastive Self-supervised Learning. In _International Conference on Learning Representations_.

---

### Review · Reviewer_2s8b · 2024-10-17

**Summary Of Contributions:**

This paper introduces a self-supervised learning framework that leverages structured data augmentations to separate the "content" and "style" attributes of data. While existing work in self-supervised learning learns to be invariant to style, essentially discarding these signals, this work proposes to disentangle the different contributors of style from one another (and from the content). The reasoning for doing so is that during pre-training we frequently don't have a lot of information about the downstream task of interest, therefore discarding certain style features might end up hurting performance.

**Audience:**

Yes

**Broader Impact Concerns:**

No broader impact concerns.

**Claims And Evidence:**

No

**Requested Changes:**

I have a few questions that I think can serve as a guide to design more experiments to showcase the effectiveness of the method:

- The work propose to model different style transformations, however that can again be hard to configure before the downstream task and requires extensive domain knowledge. It can also lead to many different transformations, so what happens when many transformations exist? Are the content embeddings trivial? It would be interesting to see how performance is affected when the number of embeddings/transformations changes, so you can argue that a priori domain knowledge is not necessary.
- In the discussion for the post-projector, it is stated that the boosted performance implies that more style information has been kept. I don’t think that can be implied from the performance. Having an objective way to quantify that more style information has been kept would help the paper.

Some more comments that I would like to see addressed/answered:

- It is very strong to assume the transformations won’t share any parameters (Section 3.1). This implies that for discrete transformations (where the probability of shared parameters through sampling is high) would not fit the framework.
- The losses of (3.2) and (3.3) are never actually defined. In the experimental section it may be inferred that they refer to the corresponding ones for SimCLR and VICReg, but that is never explicitly stated. Also in Table 1, is $n$ referring to the different transformations ($m$ was used before)? It’s not defined, and it’s very confusing. The notation used for the loss functions (sets) is different from the one in Table 1 (matrices) and this is discussed in the next page anyway, currently it just causes confusion.
- (4.5) and (4.6) are both really strong assumptions. Also, assuming $d_m=1$ is strong and it is not discussed at all at the discussion of Theorem 4.2. Overall I’m a bit confused with the theorem. I understand the merit of having simplified theoretical results to motivate a method, help in understanding, but in this case what is the benefit?  A2 and A3 don’t hold in practice and an identifiability result (that is not evaluated through simulations too) is not adding to the work, in my opinion.
- Overall the experiments of 5.1 and 5.2 are not clear. It’s not clear what the negative pairs are, and why these are two sections to begin with, since the setup is identical but simply evaluated on a different dataset. What augmentations are used and what is “augmentation strength” is never defined.

A final, minor comment, but Section 4.1 has missing assumptions. $f$ needs to be differentiable, otherwise learning the latent mapping through gradient methods is infeasible. Lookin at von Kugelgen, it seems that is an assumption they explicitly make by assuming smoothness, and the authors’ do explicitly assume that in Theorem 4.2 since $f$ is a diffeomorphic, however at 4.1 this is not communicated.

**Strengths And Weaknesses:**

**Strengths**

I find the proposed framework to be intuitive: it seems natural to model explicitly the different transformations that we can encounter, instead of lumping them all together. I also find the argument of actually keeping the information about these transformations, instead of discarding them promising and intuitive. The presentation of the work is clear and easy to follow, and the manuscript is overall very well written.

**Weaknesses**

There are some weaknesses in the work. Bluntly, the greatest one is that the experiments are fairly lackluster, and most importantly that the method actually performs worse than existing methods. Not to say that every single method needs to improve SotA, however I find that the benefits of the framework need to be identified and showcased, and the experiments that are presented here, in my opinion, fail to achieve that (I will include some suggestions in the next section).

Specifically, it is unclear how 5.1 is different from 5.2 and what these sections are overall contributing to the work. The whole premise of the work is to disentangle the different styles, and these sections consider only the content. These experiments (which make up the majority of the experimental section) showcase that SimCLR with an adjustable learning rate introduces more invariance, which seems relatively disconnected from the overall thesis of the work.

Finally, the ImageNet experiment does not currently support the effectiveness of the method. The overall performance barely changes with the proposed method when fine-tuning is used, and significantly deteriorates when trained from scratch. Even in the case of the post-projector, in some cases an improvement is not observed. I would also argue that training from scratch doesn’t seem to boost performance even more, considering it completely fails for VICReg.

---

> ### Author Response · Authors · 2024-10-31
> **Response to Reviewer 2s8b (part 1)**
>
> We sincerely thank the reviewer for their time and thorough review. We address the concerns and requested changes below.
>
> ## Weaknesses
> > _“it is unclear how 5.1 is different from 5.2 and what these sections are overall contributing to the work. The whole premise of the work is to disentangle the different styles, and these sections consider only the content. These experiments (which make up the majority of the experimental section) showcase that SimCLR with an adjustable learning rate introduces more invariance, which seems relatively disconnected from the overall thesis of the work.”_
>
> The importance and relevance of these sections is as follows:
> - In standard SSL methods, the relative weighting of invariance and entropy terms (see Table 1) is chosen to maximize performance on a particular task (e.g., ImageNet Top1 accuracy). In effect, this determines how much style to keep.
> - In contrast, as we seek to recover style information in a _separate_ embedding space, we can instead choose this relative weighting ($\lambda_0$ in Eq. 3.2) to _completely remove style from the content embedding space_.
> - Moreover, demonstrating that we can do so empirically is the first and most important step towards disentangled embedding spaces, since all other embedding spaces have the same invariance-entropy trade-offs but with different transformations (see Eq. (3.2)).
> - Thus, the experiments of Sections 5.1 and 5.2 show that:
>   - i. we can completely and consistently remove all of this “style” (by appropriately setting $\lambda_0$ in Eq. (3.2)), leaving only the information that we wish to capture in that space; and
>   - ii. many factors affect the amount of unwanted “style” retained in a particular embedding space, e.g., the size of the embedding, the strength of the augmentations, etc.
>
> We also note that there may be a crucial misunderstanding here: we use SimCLR with an adjustable invariance-entropy tradeoff ($\lambda_0$ in Eq. 3.2) rather than “an adjustable learning rate”.
>
>
> > _“the ImageNet experiment does not currently support the effectiveness of the method. The overall performance barely changes with the proposed method when fine-tuning is used, and significantly deteriorates when trained from scratch. Even in the case of the post-projector, in some cases an improvement is not observed.”_
>
> While the pre-projector results did not show a significant improvement, we note that:
> - **The post-projector results show a clear improvement** (all fine-tuned models improve, and “SimCLR + Ours-Scratch” performs best on 3 of 4 downstream-task groups) **and this supports the effectiveness of our method in keeping more style information in the space which is optimized (i.e., the embedding space $\mathbf{z}$)**.
> - The **pre-projector results** provide the community with more evidence of a large performance disparity between the pre-projector $\mathbf{h}$ and post-projector $\mathbf{z}$, **raising further questions about the role of the projector in SSL—particularly in regard to style retention**. With the projector’s role already an active area of research [1-4], we believe that our results have clear value for the community and motivate further research into the role of the projector for style retention.
> - As noted by Reviewer hV11, we tried to be extremely honest about these results and not overclaim. However, in response to the quoted comment above, **we have further toned down the empirical claims of the paper to ensure that they are fully supported by the experimental evidence** (see the general response above for a list of changes). We hope that the reviewer now agrees that all empirical claims are fully supported by the experimental evidence. However, if we have missed anything, or if there are any remaining concerns/suggestions, please let us know.
>
>
> ## Requested changes
> > _“The work propose to model different style transformations, however that can again be hard to configure before the downstream task and requires extensive domain knowledge. It can also lead to many different transformations, so what happens when many transformations exist? Are the content embeddings trivial? It would be interesting to see how performance is affected when the number of embeddings/transformations changes, so you can argue that a priori domain knowledge is not necessary.”_
>
> **Domain knowledge:** It is important to distinguish between (a) domain knowledge of the data structures and (b) _a priori_ knowledge of the downstream task. We argue that (b) is not necessary, since it is often unavailable and goes against the promise of large-scale (“universal”) pretraining. However, we never argue that (a) is not necessary—self-supervised learning is built on knowledge of data structures (e.g., predicting one part of the input from another, or using image transformations that only affect appearance attributes).
>
> (... continued below)

---

> > ### Author Response · Authors · 2024-10-31
> > **Response to Reviewer 2s8b (part 2)**
> >
> > **Many transformations:** When many transformations are available from domain knowledge, we suggest grouping them (e.g., into appearance and spatial transformations), as we did for the ImageNet experiments. For the content embedding, we agree that it’s interesting to think about what happens in the limit of many style transformations (e.g., do content embeddings become trivial?), but believe that this lies outside the scope of the current work.
> >
> > > _“In the discussion for the post-projector, it is stated that the boosted performance implies that more style information has been kept. I don’t think that can be implied from the performance. Having an objective way to quantify that more style information has been kept would help the paper.”_
> >
> > For the synthetic datasets of Sections 5.1 and 5.2, we indeed use objective measures of style retention. However, on real-world data like ImageNet, where there are no ground-truth labels for style features, we cannot do this. Instead, we adopt the standard approach/proxy [1-3]: find downstream tasks that heavily rely on one particular feature (e.g., color or spatial information), and use this as a proxy for the amount of information retained about that feature. One alternative is to use transformation invariance [3], but this only tells us how sensitive our embedding is to changes, not how much information has been captured.
> >
> > > _“It is very strong to assume the transformations won’t share any parameters (Section 3.1). This implies that for discrete transformations (where the probability of shared parameters through sampling is high) would not fit the framework.”_
> >
> > In Section 3.2, we use “no shared parameters” to build intuition (so the samples of Figure 1 make sense) and to prepare for the theory of Section 4. However, in our experiments, we do indeed use some discrete transformations (e.g., flip and grayscale) that can share parameters. **We thank the reviewer for pointing this out and have updated the footnote on page 4 of the manuscript to reflect the fact that this is not required in practice for our framework.**
> >
> > > “The losses of (3.2) and (3.3) are never actually defined. In the experimental section it may be inferred that they refer to the corresponding ones for SimCLR and VICReg, but that is never explicitly stated.”
> >
> > To allow any “base” algorithm to be used, (3.2) and (3.3) are intentionally defined using general invariance and entropy terms. These general terms are connected to their concrete specifications in Table 1 (e.g., for SimCLR, VICReg and BarlowTwins) in the text that immediately follows (3.2) and (3.3). However, we take the reviewer’s point that this was never explicitly stated, so **we have now added a line after (3.2) and (3.3) in the updated manuscript**.
> >
> > > _“in Table 1, is $n$ referring to the different transformations ($m$ was used before)? It’s not defined, and it’s very confusing.”_
> >
> > $n$ refers to the number of observations (“batches of $n$ vectors”), not the different transformations $m$. **We have updated Table 1's caption to make this point more clear**.
> >
> > > _“(4.5) and (4.6) are both really strong assumptions.”_
> >
> > (4.5) actually comes without loss of generality, since any vector can be partitioned in such a way.
> >
> > (4.6), which states that each modified style latent depends only on the value of its unmodified version and not on other style variables, indeed constitutes a non-trivial assumption. However, we disagree that it is particularly strong or unreasonable, as identifiability results for multi-view data commonly rely on the stronger assumption that such factorization holds for _all_ latents (including content) [e.g., 8, 9, 10].
> >
> > > _“assuming $d_m=1$ is strong and it is not discussed at all at the discussion of Theorem 4.2.”_
> >
> > Unlike for (4.5) and (4.6), we agree this assumption is restrictive. Unfortunately, it appears to be _necessary_ for complete style disentanglement: As explained in footnote 5, univariate style components are required to invoke Lemma 2 of Brehmer et al. in Step 3 of the proof. This result links statistical to functional independence but generally does not hold in higher dimensions.
> >
> > **We have moved the footnote to the main text and expanded on this point in the discussion paragraph after Thm 4.2 in the revised manuscript**.

---

> ### Author Response · Authors · 2024-10-31
> **Reviewer 2s8b (part 3)**
>
> > _“Overall I’m a bit confused with the theorem. I understand the merit of having simplified theoretical results to motivate a method, help in understanding, but in this case what is the benefit? A2 and A3 don’t hold in practice and an identifiability result (that is not evaluated through simulations too) is not adding to the work, in my opinion.”_
>
> While our theoretical analysis does not perfectly match practical conditions, we note that:
> - This is generally true for theoretical work (e.g., i.i.d. data, Gaussianity, or linearity are typically violated on real-world data).
> - It still provides value by:
>   - **Making explicit the assumptions needed to guarantee our approach works** (in contrast to most SSL methods which come without any theoretical/identifiability results, making it unclear what assumptions and conditions are required).
>   - **Suggesting fundamental limitations for the learning goal of interest**. E.g., $A_3$ suggests complete disentanglement may be limited to univariate style components, while $A_2$ suggests that the (independent) exogenous style terms $u_m$ may sometimes be a more realistic learning target than the style variables $s_m$.
>   - **Advancing the literature on identifiable and causal representation learning**. There is a sizable research community that studies the identifiability of statistical and causal representations. This community wishes to make clear when a given learning objective is (at least in principle) _provably_ achievable. This requires considering idealized conditions with infinite data and restrictive assumptions, but the resulting necessary and sufficient conditions often yield new insights and provide us with a better understanding of when a given method can be expected to perform as intended.
>
> Thus, we must respectfully disagree with the reviewer’s opinion that our theoretical results add no value to the work.
>
>
> > _“Overall the experiments of 5.1 and 5.2 are not clear. It’s not clear what the negative pairs are, and why these are two sections to begin with, since the setup is identical but simply evaluated on a different dataset. What augmentations are used and what is “augmentation strength” is never defined.”_
>
> - **Motivation for 5.1 and 5.2:** See “Sections 5.1 and 5.2” under Weaknesses above.
> - **Negative pairs:** these are different sample/observation indices, as usual—we are using the standard SimCLR/VICReg setups but tuning the invariance-entropy trade-off $\lambda$.
> - **Augmentation strengths:** For a visual depiction, see Figure 4 in the Appendix. We have now added the exact augmentation parameters for weak, medium and strong categories to Table 4 of Appendix A.1.
>
> > _“A final, minor comment, but Section 4.1 has missing assumptions. $f$ needs to be differentiable, otherwise learning the latent mapping through gradient methods is infeasible. Lookin at von Kugelgen, it seems that is an assumption they explicitly make by assuming smoothness, and the authors’ do explicitly assume that in Theorem 4.2 since $f$ is a diffeomorphic, however at 4.1 this is not communicated.”_
>
> In fact, $f$ need not be differentiable for the statements in Sec. 4.1 to hold. Differentiability of $f$ (and $f^{-1}$) is only required in (the proof of) Thm. 4.2 for the transformation of densities by these functions to be valid. As noted by the reviewer, differentiability is assumed in $A_1$ (“$f$ is diffeomorphic onto its image”). For ease of presentation, we only included the differentiability assumption in the Theorem where it is actually needed, but we are happy to include it in Sec 4.1 if the reviewer thinks this would improve clarity.
>
> [1] Xiao, T., Wang, X., Efros, A. A., & Darrell, T. (2021). What Should Not Be Contrastive in Contrastive Learning. In _International Conference on Learning Representations_.
>
> [2] Lee, H., Lee, K., Lee, K., Lee, H., & Shin, J. (2021). Improving transferability of representations via augmentation-aware self-supervision. In _Advances in Neural Information Processing Systems 34_, 17710-17722.
>
> [3] Ericsson, L., Gouk, H., & Hospedales, T. (2022). Why do self-supervised models transfer? on the impact of invariance on downstream tasks. In _The 33rd British Machine Vision Conference_ (p. 509).
>
> [4] Bordes, F., Balestriero, R., Garrido, Q., Bardes, A., & Vincent, P. (2024). Guillotine Regularization: Why removing layers is needed to improve generalization in Self-Supervised Learning. In _Transactions on Machine Learning Research_.
>
> [5] Jing, L., Vincent, P., LeCun, Y., & Tian, Y. (2022). Understanding Dimensional Collapse in Contrastive Self-supervised Learning. In _International Conference on Learning Representations_.
>
> [6] Gupta, K., Ajanthan, T., Hengel, A. V. D., & Gould, S. (2022). Understanding and improving the role of projection head in self-supervised learning. _arXiv preprint arXiv:2212.11491_.

---

> > ### Author Response · Authors · 2024-10-31
> > **Reviewer 2s8b (part 4)**
> >
> > [7] Xue, Y., Gan, E., Ni, J., Joshi, S., & Mirzasoleiman, B. (2024). Investigating the Benefits of Projection Head for Representation Learning. In _The Twelfth International Conference on Learning Representations_.
> >
> > [8] Zimmermann, R. S., Sharma, Y., Schneider, S., Bethge, M., & Brendel, W. (2021). Contrastive learning inverts the data generating process. In _International Conference on Machine Learning_ (pp. 12979-12990).
> >
> > [9] Gresele, L., Rubenstein, P. K., Mehrjou, A., Locatello, F., & Schölkopf, B. (2020). The incomplete rosetta stone problem: Identifiability results for multi-view nonlinear ICA. In _Uncertainty in Artificial Intelligence_ (pp. 217-227).
> >
> > [10] Klindt, D. A., Schott, L., Sharma, Y., Ustyuzhaninov, I., Brendel, W., Bethge, M., & Paiton, D. (2021). Towards Nonlinear Disentanglement in Natural Data with Temporal Sparse Coding. In _International Conference on Learning Representations_.

---

### Author Response · Authors · 2024-10-31
**General response to all reviewers**

We sincerely thank all reviewers for their time and thoughtful feedback.

We were pleased to read that reviewers found our proposed framework _“natural”_ and _“intuitive”_ (2s8b), the experiments _“well-motivated”_, _“sensible”_ and _“honest”_ (hV11), and the paper _“very well written”_ (2s8b, hV11).


### ImageNet results
The main concern of reviewers surrounds the ImageNet results. In particular, the primary concern of Reviewers 2s8b and H2K4 is that, while our method improves post-projector performance on downstream tasks, it does not significantly improve the pre-projector performance. As noted by Reviewer hV11, we are extremely honest about this, as we believe that:
- the **post-projector results** are sufficient to demonstrate the ability of our method to retain style information in the space which is optimized (i.e., the embedding space $\mathcal{Z}$).
- the **pre-projector results** provide the community with more evidence of a large performance disparity between the pre-projector $\mathbf{h}$ and post-projector $\mathbf{z}$, raising further questions about the role of the projector in SSL—_particularly in regard to style retention_. With the projector’s role already an active area of research [1-4], we believe that our results have clear value for the community and motivate further research into the role of the projector for style retention.

To ensure that this is clear, we have made the following updates to the manuscript in order to tone down any claims about improving downstream performance:
- _Introduction:_ Under structure and contributions, we have adjusted our statements about the experiments to be careful/conservative. In particular, we have emphasized that:
  - (i) improvements in downstream performance and in style retention come with the post-projector embedding $\mathbf{z}$; and
  - (ii) there remains a large performance disparity between $\mathbf{z}$ and the pre-projector representation $\mathbf{h}$.
- _Experiments:_ In the introduction of Section 5 and Section 5.3, we have again adjusted our statements in a similar manner.
- _Conclusion:_ Similar adjustments.

We believe that these updates address the reviewers' concerns, **ensuring that all empirical claims are fully supported by the experimental evidence**. However, if we have missed anything, we welcome comments and suggestions from the reviewers.


### Other
The remaining reviewer-specific concerns stem from items that may have been unclear or misunderstood, e.g., Reviewer 2s8b found the purpose of Sections 5.1 and 5.2 unclear, while Reviewer H2K4 thought that we choose to keep one of $M$ style variables (rather than all $M$). We believe that we have thoroughly addressed these points in our responses to the reviewers and the revised manuscript.


### Discussion
If the reviewers agree that their concerns have been addressed, we kindly ask that they take this into account when considering score adjustments. If the reviewers have any further questions or comments, we are very happy to respond during the discussion phase.


[1] Bordes, F., Balestriero, R., Garrido, Q., Bardes, A., & Vincent, P. (2024). Guillotine Regularization: Why removing layers is needed to improve generalization in Self-Supervised Learning. In _Transactions on Machine Learning Research_.

[2] Jing, L., Vincent, P., LeCun, Y., & Tian, Y. (2022). Understanding Dimensional Collapse in Contrastive Self-supervised Learning. In _International Conference on Learning Representations_.

[3] Gupta, K., Ajanthan, T., Hengel, A. V. D., & Gould, S. (2022). Understanding and improving the role of projection head in self-supervised learning. _arXiv preprint arXiv:2212.11491_.

[4] Xue, Y., Gan, E., Ni, J., Joshi, S., & Mirzasoleiman, B. (2024). Investigating the Benefits of Projection Head for Representation Learning. In _International Conference on Learning Representations_.

---

### Decision · Action_Editor_XvPo · 2024-11-24

**Recommendation:** Reject

**Comment:**

Overall, the paper proposes a promising extension of existing work, but additional and careful experimental analysis is needed to support the main claim about the benefits and the scalability of disentangling style variables in self-supervised learning. Given this, I recommend rejecting the paper, but I encourage the authors to revise and resubmit their work.

**Audience:**

The work targets the self-supervised learning community. While the overall idea of disentangling 'style' variables from 'content' is reasonable, its benefits are not convincingly demonstrated, limiting its appeal and potential adoption by the community.

**Claims And Evidence:**

All reviewers praised the clarity of the writing and the simplicity of the idea behind identifying and disentangling different style variables. However, two out of the three reviewers expressed concerns about the scalability of the proposed approach to real-world data. These concerns included the ambiguity of the concept of 'style' in self-supervised learning, the requirement for domain knowledge to define the style variables, and the lack of convincing experimental evidence—especially with respect to ImageNet data. Regarding the latter point, both reviewers were particularly concerned about the absence of a clear improvement over existing methods and the significant performance gap (around 20%) between the pre- and post-projector results, which undermined the credibility of the claim that more style information is retained.

**Resubmission Of Major Revision:**

The authors may consider submitting a major revision at a later time.